# Crystal structures reveal catalytic and regulatory mechanisms of the dual-specificity ubiquitin/FAT10 E1 enzyme Uba6

Lingmin Yuan[1,8], Fei Gao[1,2,8], Zongyang Lv[1], Digant Nayak [1], Anindita Nayak [1], Priscila dos Santos Bury[1], Kristin E. Cano[1], Lijia Jia[1], Natalia Oleinik[3], Firdevs Cansu Atilgan [3], Besim Ogretmen [3], Katelyn M. Williams[4], Christopher Davies[5], Farid El Oualid [6], Elizabeth V. Wasmuth[1,7] & Shaun K. Olsen [1] ✉

The E1 enzyme Uba6 initiates signal transduction by activating ubiquitin and the ubiquitin-like protein FAT10 in a two-step process involving sequential catalysis of adenylation and thioester bond formation. To gain mechanistic insights into these processes, we determined the crystal structure of a human Uba6/ubiquitin complex. Two distinct architectures of the complex are observed: one in which Uba6 adopts an open conformation with the active site configured for catalysis of adenylation, and a second drastically different closed conformation in which the adenylation active site is disassembled and reconfigured for catalysis of thioester bond formation. Surprisingly, an inositol hexakisphosphate (InsP6) molecule binds to a previously unidentified allosteric site on Uba6. Our structural, biochemical, and biophysical data indicate that InsP6 allosterically inhibits Uba6 activity by altering interconversion of the open and closed conformations of Uba6 while also enhancing its stability. In addition to revealing the molecular mechanisms of catalysis by Uba6 and allosteric regulation of its activities, our structures provide a framework for developing Uba6-specific inhibitors and raise the possibility of allosteric regulation of other E1s by naturally occurring cellular metabolites.

Post-translational modification (PTM) of proteins by ubiquitin (Ub) and Ub-like proteins (Ubls) underpins the regulation of fundamental cellular processes including cell cycle control, DNA repair, signal transduction, and immunity[1]. Ub/Ubl conjugation requires the sequential interactions and activities of parallel cascades of enzymes comprising E1, E2, and most often E3s, which together activate, shuttle, and ligate Ub/Ubl to target proteins[2]. The two human Ub E1s, Uba1 and Uba6, function as gatekeepers of the Ub-conjugation cascade by coupling ATP hydrolysis (adenylation) with the formation of a high-energy E1-Ub thioester linkage (thiolation). This is then followed by the transfer of activated Ub to distinct repertoires of cognate E2 enzymes (transthioesterification)[3–9]. While Uba1 is specific for Ub activation, Uba6 exhibits dual specificity for Ub and FAT10 [5,8], the latter being a Ubl involved in mitotic progression[10], immunity[11–14], and implicated in cancer[15–21].

[1]Department of Biochemistry & Structural Biology, University of Texas Health Science Center at San Antonio, San Antonio, TX 78229, USA. [2]Department of Research & Development, Beijing IPE Center for Clinical Laboratory CO, Beijing 100176, China. [3]Department of Biochemistry & Molecular Biology and Hollings Cancer Center, Medical University of South Carolina, Charleston, SC 29425, USA. [4]Department of Pediatrics, Johns Hopkins University School of Medicine, Baltimore, MD 21287, USA. [5]Department of Biochemistry & Molecular Biology, University of South Alabama, Mobile, AL 36688, USA. [6]UbiQ Bio B.V., Science Park 408, 1098 XH Amsterdam, The Netherlands. [7]Human Oncology and Pathogenesis Program, Memorial Sloan Kettering Cancer Center, New York, NY 10065, USA. [8]These authors contributed equally: Lingmin Yuan, Fei Gao. ✉e-mail: olsens@uthscsa.edu

While Uba6 has eluded structural characterization to date, a wealth of studies have shown that Uba1 is a large multidomain enzyme in which each domain plays a distinct functional role during catalysis of Ub adenylation[22,23], E1-Ub thioester bond formation[24–26], and E1-E2-Ub transthioesterification[27–30]. Active and inactive adenylation domains (AAD and IAD, respectively) are responsible for the initial recruitment of Ub and also harbor the catalytic machinery for adenylation of the *C*-terminus of Ub in the first step of Ub activation. The E1 Cys domain is arranged as two globular half domains termed the first and second catalytic cysteine half domains (FCCH and SCCH, respectively). The FCCH domain plays a role in Ub recognition and the SCCH domain harboring the catalytic cysteine residue involved in Ub thioester bond formation during the second step of Ub activation. Recent studies have revealed that E1s undergo large conformational changes that are required for adenylation and thioester bond formation[24,25,31,32]. Studies of Uba1[25], as well as the E1 for the Ubl SUMO, have shown that adenylation occurs with the E1 in an "open" conformation and that thioester bond formation involves a -130° rotation (or "closure") of the SCCH domain that transits the catalytic cysteine into the active site[31]. Lastly, the ubiquitin-fold domain (UFD) is involved in the molecular recognition of E2 and subsequent transfer of Ub from the E1 to E2 catalytic cysteine[26–30]. Uba6 is predicted to harbor a similar domain organization as Uba1 based on sequence analysis[5]. In the absence of structural data, however, the molecular mechanisms by which Uba6 activates Ub and FAT10 are unknown. Furthermore, while nearly all Uba1 in proliferating mammalian cells is in the activated Uba1-Ub state, Uba6 is only 50% activated under similar conditions[6,33]. The molecular basis for this difference remains to be determined.

Given the crucial role Ub conjugation plays in the regulation of myriad cellular processes in humans, it is perhaps not surprising that the enzymatic machinery responsible for Ub conjugation is itself subject to multiple layers of regulatory mechanisms that allow for fine-tuning and modulation of its function[34,35]. E1, E2, and E3 enzymes functioning in multiple Ub/Ubl pathways, as well as the Ub/Ubls substrates themselves, are regulated by PTMs, such as phosphorylation, acetylation, deamidation, ADP ribosylation, ubiquitination, SUMOylation and NEDDylation[34,35]. In addition to PTMs, several naturally occurring small molecules are known to regulate Ub/Ubl pathways. For example, members of the largest family of E3 Ub ligases, the cullin-RING E3 Ub ligases (CRLs), are negatively regulated by inositol hexakisphosphate (InsP6) and other inositol phosphates[36–39]. Specifically, CRLs are deactivated by COP9 signalosome-mediated deneddylation and InsP6 serves as a cofactor that promotes the CRL/COP9 interaction[38,39]. InsP6 and InsP5 have also been revealed to directly interact with the substrate adapter subunits of the plant CRLs SCF$^{TIR1}$ and SCF$^{COI1}$, playing a structural role in supporting TIR1 and COI1 function as auxin and jasmonate receptors, respectively[36,37]. Considering the steady pace of discovery of PTM and small molecule-mediated regulatory mechanisms of Ub signaling and the relative paucity of studies focused on Uba6, mechanisms that govern the activity of Uba6 remain unknown.

To gain insights into the molecular basis by which Uba6 catalyzes the activation of Ub/FAT10, we determined the 2.25 Å crystal structure of a human Uba6/Ub complex. The Uba6/Ub complex is observed in two drastically different conformations: an open conformation poised for adenylation and a closed complex poised for thioester bond formation. Moreover, we observed an inositol hexakisphosphate (InsP6) molecule bound to an evolutionarily conserved, highly basic pocket unique to the Uba6 SCCH domain. Contacts with InsP6 in the open conformation involve several residues and structural elements that undergo rearrangement during thioester bond formation and residues important for catalysis. We also find that InsP6 allosterically inhibits Uba6 activity by altering interconversion of the open and closed conformations of Uba6 while also enhancing its stability. Collectively, our studies reveal the structural basis for Uba6 catalytic activities and

unexpectedly unveil a unique mechanism of allosteric regulation by a naturally occurring cellular metabolite that exploits specific mechanistic features of Uba6.

## Results

### Overall structure of Uba6/Ub in open and closed conformations

To elucidate the molecular mechanisms by which Uba6 activates Ub/FAT10, we sought to determine a crystal structure of a human Uba6/Ub complex. A catalytic cysteine to alanine mutant of human Uba6 (C625A) capable of catalyzing adenylation but not thioester bond formation (thiolation) (Fig. 1a) was used in our structural studies to reduce sample heterogeneity and facilitate crystallization. Insect cell-derived recombinant human Uba6$^{C625A}$ was used in our structural studies as protein yields were significantly higher compared to expression in bacteria. Prior to conducting crystallization trials, human Uba6$^{C625A}$ was mixed with Ub and ATP·Mg$^{2+}$ to promote adenylation, and diffraction-quality crystals harboring two complexes in the asymmetric unit (AU) were obtained. After extensive screening and refinement, a structure of the human Uba6$^{C625A}$/Ub complex was determined to 2.25 Å resolution, with R/R$_{free}$ values of 0.166/0.206 and excellent geometry (Supplementary Table 1). Ub from both copies of the human Uba6$^{C625A}$/Ub complex in the AU exhibited strong and continuous electron density at their *C*-termini, consistent with the Ub-adenylate product. By contrast, electron density corresponding to the pyrophosphate (PPi) leaving group and Mg$^{2+}$ was absent (Supplementary Fig. 1a). Thus, the complexes represent the product complex of adenylation, after the release of PPI and Mg$^{2+}$.

Remarkably, the two copies of the human Uba6$^{C625A}$/Ub-adenylate complex in the crystallographic AU exhibit drastically different architectures relative to one another (Fig. 1b, c). In one copy, the adenylation active site is completely ordered and the SCCH domain adopts an "open" conformation where the catalytic cysteine (C625A) is more than 35 Å from the glycyl-phosphate linkage of Ub-AMP that is nucleophilically attacked during thioester bond formation (Fig. 1b). We hereafter refer to this adenylation-competent conformation of the human Uba6$^{C625A}$/Ub-adenylate complex as Uba6$^{OPEN}$/Ub-AMP. In the other copy of the complex in the AU, the adenylation active site is partially disassembled and the SCCH domain adopts a "closed" conformation exhibiting a 136° rigid body rotation (relative to Uba6$^{OPEN}$/Ub-AMP). This places the β-carbon of the catalytic cysteine (C625A) -4 Å from the glycyl-phosphate linkage of Ub-AMP and brings additional structural elements important for thioester bond formation into the active site (Fig. 1c). We hereafter refer to this thiolation-competent conformation of the complex as Uba6$^{CLOSED}$/Ub-AMP. Further comparison shows that the UFD of the Uba6$^{CLOSED}$/Ub-AMP structure undergoes a 22° rigid body rotation towards the SCCH domain, relative to the Uba6$^{OPEN}$/Ub-AMP structure (Fig. 1b, c). These structural changes result in the canyon between the UFD and SCCH domain reducing in length from -35 Å in the Uba6$^{OPEN}$/Ub-AMP to -24 Å in the Uba6$^{CLOSED}$/Ub-AMP structure (Fig. 1c). An additional surprise in the Uba6$^{OPEN}$/Ub-AMP structure was the presence of strong electron density corresponding to an inositol hexakisphosphate (InsP6) molecule located in the SCCH domain (Fig. 1b). Lastly, an ordered loop region within the SCCH domain that occludes the catalytic cysteine in the Uba6$^{OPEN}$/Ub-AMP structure (termed the "cys cap") becomes disordered in the Uba6$^{CLOSED}$/Ub-AMP structure. How the observed conformational changes and binding by InsP6 are mechanistically and functionally related to Uba6 activity is discussed below.

Overall, the Uba6/Ub interaction is similar to that of Uba1/Ub[22,23] and to gain insights into the dual specificity of Uba6 for Ub and FAT10, we created a Uba6/FAT10 model by docking FAT10 (PDB: 6GF2)[40] onto Ub from the Uba6/Ub structure (Supplementary Fig. 1d, e). The Uba6/FAT10 model and sequence analysis reveal that FAT10 harbors only 31% identity at Ub positions that interact with Uba6 (Supplementary Fig. 1f). Among the many regions of divergence between Ub and

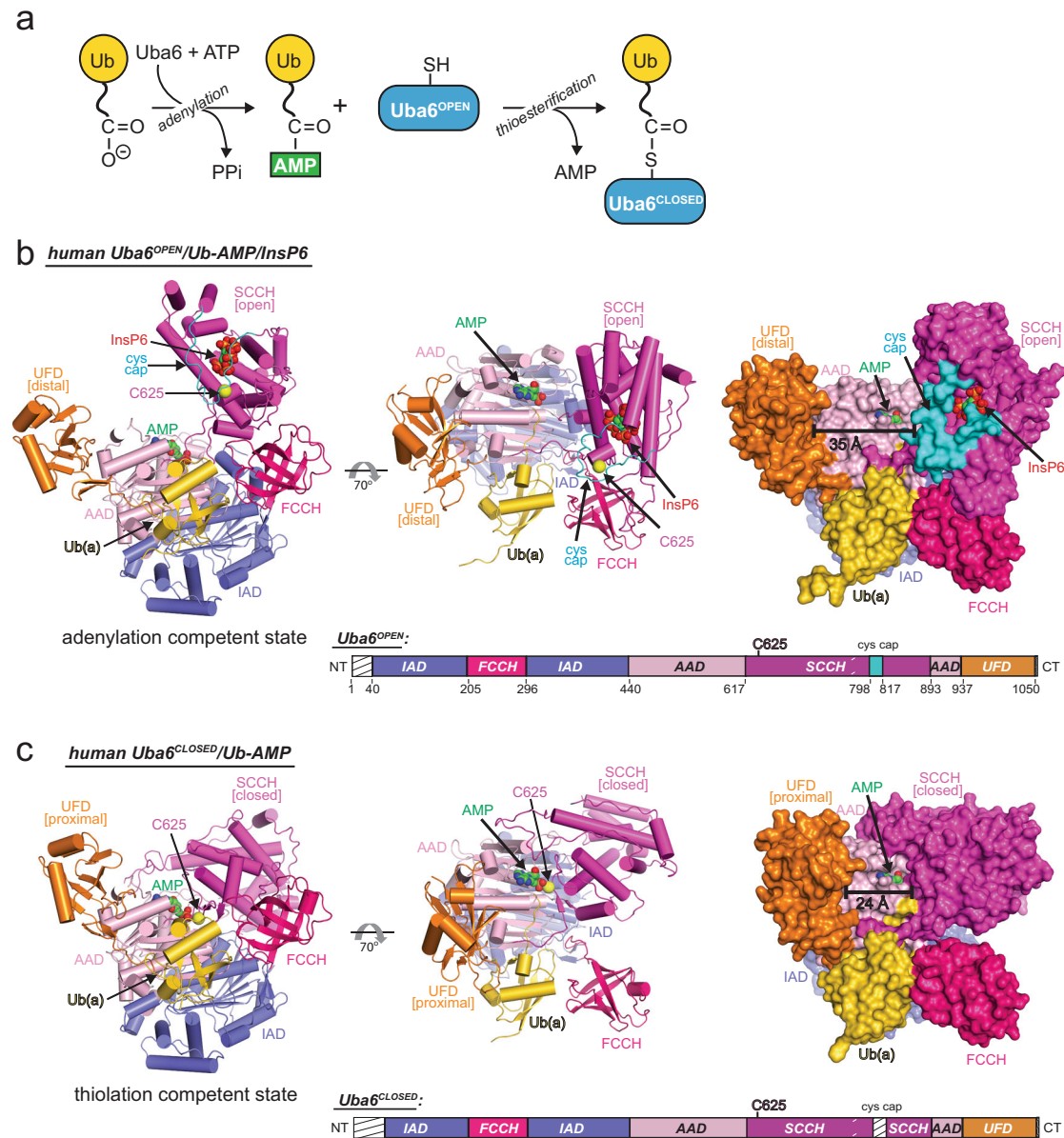

**Fig. 1 | Overall architecture of Uba6/Ub-AMP complexes in open and closed states. a** Schematic representation of Uba6-Ub thioester formation reaction. **b** Cartoon representation of crystal structure of Uba6/Ub-AMP/InsP6 in the open E1 conformation. IAD domain is in slate; AAD is in pink; FCCH is in hotpink; SCCH is in magenta; UFD is in orange; cys cap is in cyan; and Ub(a) is in gold. The catalytic cysteine is indicated by yellow sphere. AMP and InsP6 are indicated by spheres. **c** Cartoon representation of crystal structure of hUba6/Ub-AMP in the closed E1 conformation presented as in **b**. Schematic drawings of the domain organization of Uba6 with disordered regions indicated by hatched boxes are shown at the bottom of **b** and **c**.

FAT10, the Ile44 hydrophobic patch of Ub (Leu8/Ile44/Val70), which is known to be crucial for Uba1 activation of Ub[41] and participates in a similar network of hydrophobic interactions in the Uba6/Ub structure, is not conserved in FAT10 (Supplementary Fig. 1d–f). In FAT10, the region corresponding to the Ub Ile44 patch has diverged to be much more polar and comprises Ser95, Thr132, and Ala159. Thus, the same network of functionally important hydrophobic contacts that Ub participates in with Uba1 and Uba6 is not possible for FAT10. In addition to differences at many other Ub residues that interact with Uba6 in the Uba6/Ub structure, FAT10 has a two-residue insertion in the β1-β2 loop which is in proximity to the AAD of Uba6 where it may participate in unique interactions (Supplementary Fig. 1d–f) and a unique "CYCI" motif at the C-terminus that plays a role in specificity[42]. Lastly, FAT10 differs from Ub in that it harbors two tandem Ub-like domains (NTD and CTD) connected by a linker region (Supplementary Fig. 1e) that

previous studies have demonstrated is crucial for activation of FAT10 by Uba6[40] despite being situated quite far from the surface of the enzyme. Precisely how the aforementioned differences in the FAT10 CTD affect its mode of binding to Uba6, how the NTD-CTD linker and possibly the NTD itself might play a role in Uba6 binding, and whether these differences may compensate for the lack of the hydrophobic Ile44 patch in FAT10 will await the determination of a Uba6/FAT10 structure.

**Uba6 domain alternation**

The SCCH domain is tethered to the adenylation domains of Uba6 by two flexible loops. These are termed the crossover loop, which harbors the catalytic cysteine and leads into the beginning of the SCCH domain, and the reentry loop, which connects the SCCH domain back into the adenylation domains (Fig. 2a). As noted above, the SCCH

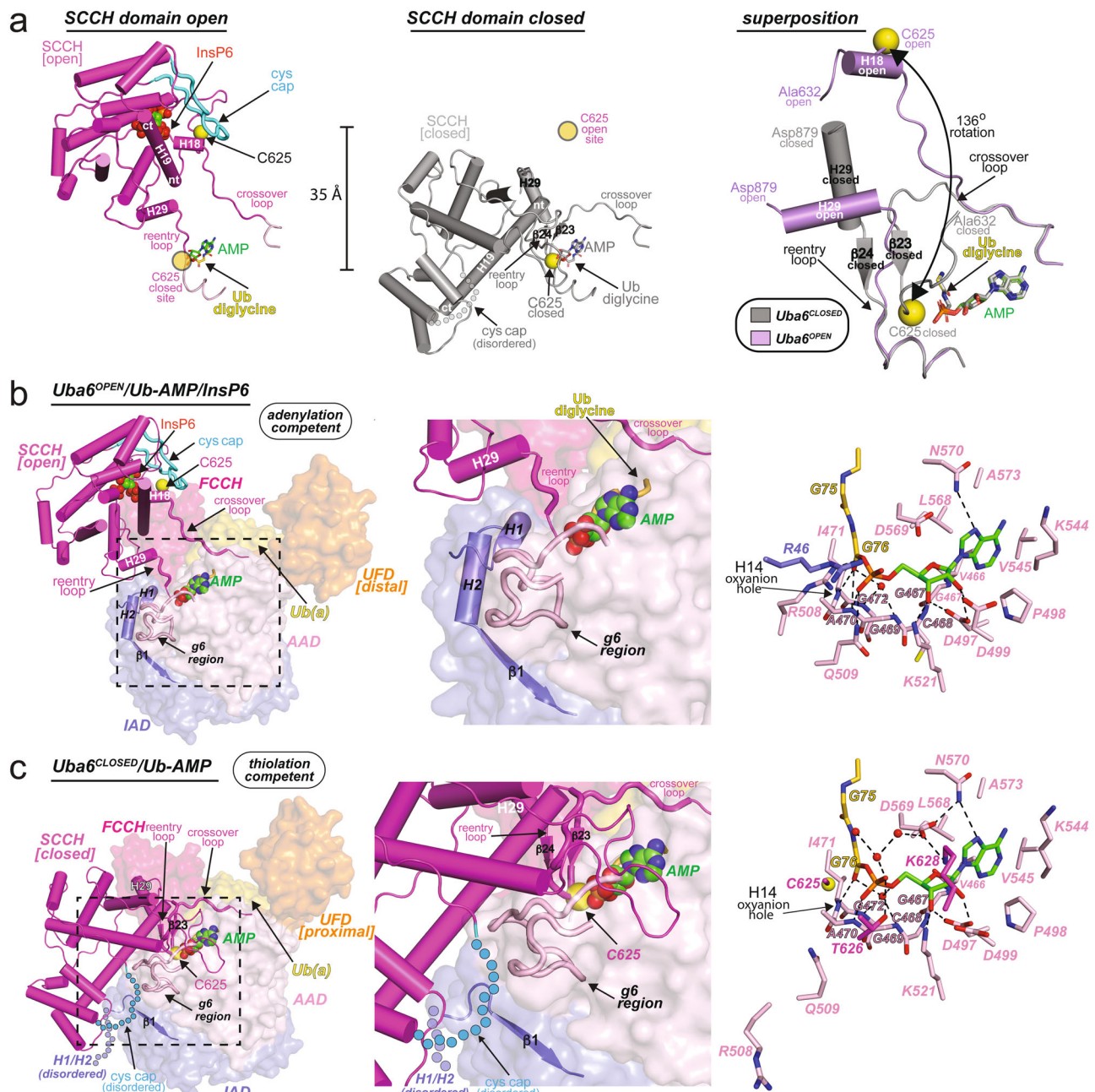

**Fig. 2 | Structural changes in Uba6 accompany transitions from adenylate to thiolation-competent states. a** Comparison of Uba6 SCCH domain in the open (adenylation active) and closed (thioester bond formation active) conformations. The adenylation domains (which serve as the rigid body of Uba6) were superimposed and the SCCH domains are shown as cartoons with the catalytic cysteines shown as spheres. The degree of rotation between each SCCH domain conformational state is indicated. To provide a frame of reference, the relative position of the catalytic cysteine in the other SCCH domain conformational states is indicated with semitransparent yellow circles and labeled accordingly in each of the panels. Selected helices of the SCCH domain are labeled and their *N*- and *C*- termini are indicated by "nt" and "ct," respectively. **b, c** Elements in the Uba6^OPEN/Ub-AMP/InsP6 (**b**, Left, Middle) and Uba6^CLOSED/Ub-AMP (**c**, Left, Middle) structures that undergo conformational changes during the transition from the adenylation to thiolation-competent states are shown as color-coded and labeled cartoons with the rest of the complex shown as surface representation. Overview of the Uba6^OPEN/Ub-AMP/ InsP6 (**b**, Right) and Uba6^CLOSED/Ub-AMP (**c**, Right) active sites with residues that contact the adenylate intermediate shown as sticks. Hydrogen bonds are shown as dashed lines and selected water molecules are shown as red spheres.

domain undergoes a 136° rotation (also termed a domain alternation) that transits the catalytic cysteine into the adenylation active site. SCCH domain alternation is accompanied by a bending of the crossover loop between residues Arg615 to Pro623 and an orthogonal bending of the reentry loop between residues Gly888 to Ala893 (Fig. 2a and Supplementary Fig. 2b). In the Uba6^OPEN/Ub-AMP structure, the catalytic cysteine is located on a short helix (H18) within the crossover loop, whereas in the Uba6^CLOSED/Ub-AMP structure, H18 melts into an extended loop. This orients the catalytic cysteine for nucleophilic

attack of the glycyl-phosphate linkage of Ub-AMP (Fig. 2a and Supplementary Fig. 1b). The altered path of the crossover loop, which positions the catalytic cysteine for thiolation, is stabilized by the formation of a short β-strand pair (β23–β24) in the Uba6^CLOSED/Ub-AMP structure, whereas the residues comprising β23-β24 are flexible loops in the Uba6^OPEN/Ub-AMP structure (Fig. 2a).

In addition to conformational changes within the crossover and reentry loops of Uba6 that accompany SCCH domain alternation, the cys cap of Uba6 (residues 798-817) transitions from an ordered state in

the Uba6$^{OPEN}$/Ub-AMP structure to a disordered state in the Uba6$^{CLOSED}$/Ub-AMP structure (Fig. 2a). In the Uba6$^{OPEN}$/Ub-AMP structure the cys cap buries the Uba6 catalytic cysteine (Fig. 2a, b), which may protect it from oxidative or chemical damage that would reduce activity. In the Uba6$^{CLOSED}$/Ub-AMP structure, SCCH domain alternation transits the cys cap into close proximity with the catalytic machinery for adenylation (Fig. 2c and Supplementary Fig. 1c). If the cys cap adopted the same conformation as in the Uba6$^{OPEN}$/Ub-AMP structure, it would engage in severe steric clashes. Thus, the disordering of the cys cap during thiolation serves two purposes: first, it increases solvent accessibility of the catalytic cysteine, thereby enhancing its reactivity, and second, it facilitates SCCH domain alternation by preventing steric clashes with the adenylation domain in the closed conformation.

## Uba6 catalytic mechanism and active site remodeling

As noted above, the active site of the Uba6$^{OPEN}$/Ub-AMP structure represents the Ub-adenylate product of the adenylation reaction after the release of the PPi leaving group and Mg$^{2+}$. With respect to the glycyl-phosphate linkage between Ub Gly76 and AMP in the Ub-adenylate, the carbonyl oxygen of Gly76 from Ub participates in a network of direct and water-mediated hydrogen bonds with the backbone nitrogens of Gly469, Ala470, Ile471, Gly472 of Uba6, and the AMP phosphate (e.g. the α-phosphate of ATP) participates in direct hydrogen bonds with Ala470, Arg508, and Gln509 (Fig. 2b). Docking of ATP·Mg$^{2+}$ from the structure of *S. pombe* Uba1 (PDB: 4II2) into the Uba6 active site, which approximates the substrate complex of the adenylation reaction, shows that Arg46 and Arg508 are positioned to interact with the γ-phosphate of ATP, Asp569 is positioned to coordinate a conserved Mg$^{2+}$ ion, and Asp499, Glu502, and Asn505 are positioned to participate in coordination of a second Mg$^{2+}$ ion (Supplementary Fig. 2a). In the Uba6$^{CLOSED}$/Ub-AMP structure, the carbonyl oxygen of Ub Gly76 participates in the same network of direct and water-mediated hydrogen bonds as observed in the Uba6$^{OPEN}$/Ub-AMP structure (Fig. 2c). The AMP phosphate participates in an equivalent hydrogen bond to the backbone nitrogen of Uba6 Ala470, and as a result of domain alternation and active site remodeling, has gained a new hydrogen bond with Thr626 while losing contacts to Arg508 and Gln509 (Fig. 2c). Lastly, a network of water-mediated hydrogen bonds takes place between the backbone nitrogen of Gly76, the AMP phosphate and adenine groups, and Uba6 Asp569, Asn570, and Lys628 that is unique to the Uba6$^{CLOSED}$/Ub-AMP structure (Fig. 2c).

Comparison of the adenylation-competent Uba6$^{OPEN}$/Ub-AMP structure with the thiolation-competent Uba6$^{CLOSED}$/Ub-AMP structure reveals an extensive network of conformational changes within the adenylation domain that are coupled with SCCH domain alternation and collectively serve to remodel the Uba6 active site to catalyze thiolation (Fig. 2b, c and Supplementary Fig. 2c). Notably, several key catalytic residues emanate from regions that undergo remodeling during the transition from the adenylation- to thiolation-competent states. This includes amino acids 499–599 from the AAD (termed the "g6 region" because it harbors the short 3$_{10}$ helix, g6), and helix H1 (residues 46–51) from the IAD (Fig. 2c and Supplementary Fig. 2c). In the Uba6$^{OPEN}$/Ub-AMP structure, the g6 region and H1 interact with each other, which helps properly position residues key for adenylation and serves as a platform for the open conformation of SCCH domain (Fig. 2b). In the thiolation-competent Uba6$^{CLOSED}$/Ub-AMP structure, the g6 region of the AAD unfolds and extends away from the AMP from the Ub-adenylate where it interacts with the SCCH domain to stabilize the closed conformation, and helices H1 and H2 of the IAD flip out of the active site in order to avoid clashes with the SCCH domain (Fig. 2c and Supplementary Fig. 2d). This results in the displacement of residues important for adenylation including Arg46, Glu502, Asn505, Arg508, and Gln509 from the active site, and replacement with residues important for thioester bond formation.

In the Uba6$^{CLOSED}$/Ub-AMP structure, the β-carbon of the C625A catalytic cysteine mutant is 4.5 Å away from the Gly76 carbonyl carbon, poised for nucleophilic attack required for thiolation (Fig. 2a, c and Supplementary Fig. 2c). When modeled as a cysteine, this distance is reduced to ~4 Å and the γ-sulfur can be placed in the proper position for reactivity with only a slight change in the path of the crossover loop. Thr626 participates in a hydrogen bond with the phosphate of AMP and Lys628 engages in a salt bridge with Asp569 from the adenylation domain (Fig. 2c), which is predicted to coordinate Mg$^{2+}$ during catalysis of adenylation (Supplementary Fig. 2a). All three of these residues (Cys625, Thr626, and Lys628) reside within helix H18 in the Uba6$^{OPEN}$/Ub-AMP structure, further affirming that melting of this helix in the Uba6$^{CLOSED}$/Ub-AMP structure is an important aspect of active site remodeling that is necessary for proper positioning of these residues for thiolation.

Collectively, our structures provide the following insights into the molecular mechanisms by which Uba6 catalyzes adenylation and thioester bond formation. The constellation of amino acids within the active site in Uba6$^{OPEN}$/Ub-AMP structure suggests that the *C*-terminal carboxylate of Ub Gly76 is coordinated for nucleophilic attack of the α-phosphate of ATP during catalysis of adenylation through hydrogen bonding and charge complementarity via interactions with Gly469, Ala470, Ile471, Gly472 at the *N*-terminal end of Uba6 helix H14. Although ATP is not present in our structure, modeling suggests that during catalysis of adenylation, the PPi leaving group is stabilized by the basic and polar residues Arg46, Asn505, Arg508, Gln509, and Lys521 (Supplementary Fig. 2a). During adenylation, in addition to stabilizing the α-phosphate that undergoes inversion during nucleophilic attack, Mg$^{2+}$ is likely to relieve electrostatic repulsion between the Ub Gly76 carboxylate and the α-phosphate of ATP. After release of the PPi leaving group, SCCH domain alternation and active site remodeling disassemble the catalytic machinery for adenylation, and reorganize it for thiolation. This drives the reaction forward while simultaneously preventing the back reaction (e.g. reformation of ATP and free Ub). In both structures, the α-phosphate and *C*-terminal carbonyl oxygen of Gly76 of Ub-AMP, which are nucleophilically attacked during adenylation and thiolation, respectively, points directly towards the *N*-terminal end of helix H14 of Uba6. In turn, this suggests that the electropositivity of the helix dipole may help stabilize the negative charge of the transition state and tetrahedral intermediates formed during both the adenylation and thiolation reactions.

## InsP6 binds a basic pocket near the Uba6 catalytic cysteine

As noted above, in our 2.25 Å Uba6$^{OPEN}$/Ub-AMP structure, we observed strong electron density within a deep pocket on the surface of the SCCH domain proximal to H18 and the Uba6 catalytic cysteine (Fig. 3a–e). Based on its six-fold symmetry and the surrounding chemical environment, we identified this density as corresponding to a molecule of InsP6 (Fig. 3d and Supplementary Fig. 3a). Since this molecule was not included in either protein or crystallization buffers, it must have co-purified with the protein following expression in insect cells. The electron density is consistent with the most stable form of InsP6, myo-InsP6, in the chair configuration with the phosphate at the 2 position being axial and the phosphates at the 1, 3, 4, 5, and 6 positions being equatorial. Analysis of the structure shows that the InsP6 molecule binds to a highly basic pocket within the globular core of the SCCH domain that is nearly fully conserved from zebrafish to humans (note that Uba6 is specific to vertebrates and sea urchin) (Fig. 3f). The axial phosphate of InsP6 participates in a network of hydrogen bonds with Lys644, Lys706, Lys709, and Tyr710, while the equatorial phosphates contact Ser647, Lys652, Lys687, Arg691, and Lys714, with Trp640 playing a structural role in organizing the binding pocket (Fig. 3a, d).

Interestingly, the cys cap and crossover loop of Uba6 harbor residues that participate in a large network of contacts to InsP6,

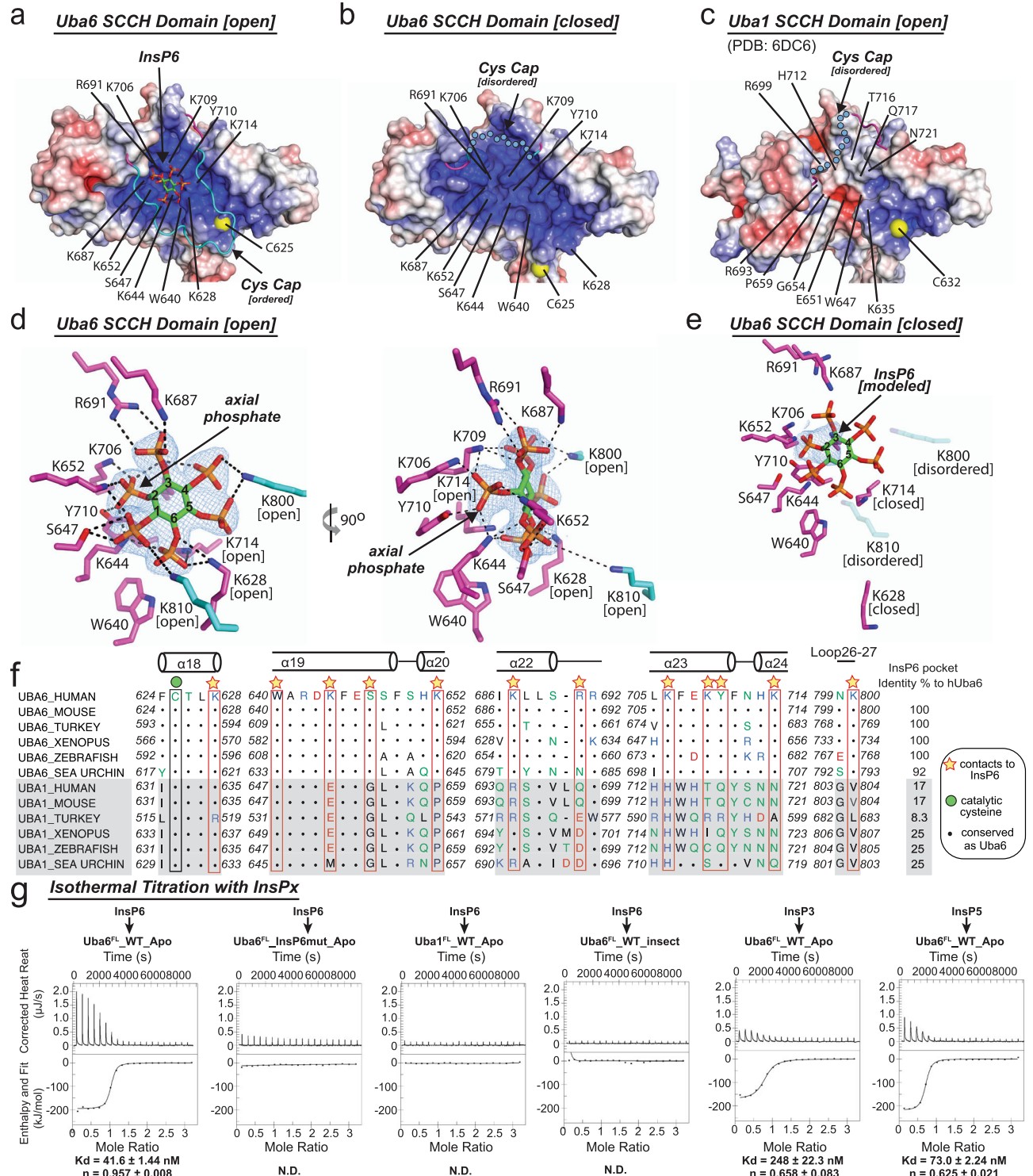

**Fig. 3 | InsP6 binds to a highly basic pocket unique to the SCCH domain of Uba6.** **a**–**c** Surface electrostatic representations of the SCCH domains of human Uba6 open (**a**) Uba6 closed (**b**) and Uba1 (**c**) structures. Uba6 residues involved in contacts to InsP6 in the Uba6 open structure and the corresponding residues from the Uba6 closed and Uba1 structures are labeled. The ordered cys cap in the Uba6 open structure is shown as cartoon and colored cyan and the disordered cys caps of Uba6 closed and Uba1 are shown as dashed cyan circles. **d** Composite omit density map (contoured at 1σ) for InsP6 in the Uba6 open structure is shown as blue mesh. InsP6 is shown as sticks with carbons (green), oxygens (red), phosphates (orange). InsP6-interacting residues are shown as sticks and hydrogen bonds are shown as

dashed lines. **e** Composite omit density map for InsP6 binding pocket in the Uba6 closed structure is shown as in **d**. InsP6 has been modeled based on the Uba6 open structure. **f** Structure-based sequence alignment of InsP6 binding region in Uba6 and Uba1 SCCH domain. Secondary structure for Uba6 is shown above. Residues important for InsP6 binding are highlighted by yellow stars. **g** Isothermal titration calorimetry data for interactions between the indicated Uba6 and Uba1 variants with the indicated inositol phosphate. Experiments were performed in triplicate and upper panels show raw power data and lower panels show fits of the data to standard binding equations using NanoAnalyze software (TA instruments). Throughout the figure, "Apo" labels refer to *E. coli*-derived material.

outside of the basic pocket on the globular SCCH domain (Fig. 3a, d). The cys cap covers the InsP6 molecule and contributes two basic residues, Lys800 and Lys810, that participate in direct and water-mediated contacts with three equatorial phosphates of InsP6 (Fig. 3a, d). In addition, Lys628, which is located in proximity to the catalytic cysteine on helix H18 within the crossover loop of Uba6 also contacts two equatorial phosphates of InsP6. Notably, the network of contacts between the cys cap, the crossover loop, and InsP6 cannot take place in the Uba6$^{CLOSED}$/Ub-AMP structure, as the cys cap becomes disordered and the crossover loop takes a drastically different path during SCCH domain alternation and active site remodeling that accompany thiolation (Fig. 2a, c). Consistent with this, we observe only weak electron density corresponding to the axial and equatorial phosphates of InsP6 in the Uba6$^{CLOSED}$/Ub-AMP structure, thus InsP6 was not included in the final model (Fig. 3e). Docking InsP6 onto the SCCH domain of the Uba6$^{CLOSED}$/Ub-AMP structure shows that there are no major steric clashes with Uba6 (Supplementary Fig. 3b), which together with the above data suggests a low-affinity binding site for InsP6 in the closed conformation.

While the residues involved in contacts to InsP6 in the Uba6$^{OPEN}$/Ub-AMP structure exhibit 92–100% identity from sea urchins to humans, the same comparison reveals that Uba1 exhibits only 8–25% identity to human Uba6 in this region with many of the differences involving charged residues (Fig. 3f). Consequently, the equivalent region in Uba1 has markedly different structural features and surface electrostatics compared to Uba6 (Fig. 3c). This suggests that the ability to interact with InsP6 has evolved as a feature specific to Uba6 but not Uba1. To test this hypothesis, we purified Uba6 and Uba1 using *E. coli* and measured affinities for InsP6 using isothermal titration calorimetry (ITC). *E. coli* was used because this organism does not produce inositol phosphates or inositol phospholipids and therefore InsP6 would not co-purify with the proteins. The results show that while bacterially derived Uba6 binds InsP6 with a $K_D$ of ~41 nM, Uba1 fails to exhibit detectable binding under the same conditions (Fig. 3g). We also tested the ability of Uba6 to bind to the signaling inositol phosphates InsP3 (D-myo-Inositol 1,4,5-trisphosphate) and InsP5 (D-myo-Inositol 1,3,4,5,6-pentakisphosphate), which are both precursors of InsP6, and found that they were also able to interact with Uba6 albeit with higher $K_D$ values of 248 and 73 nM, respectively (Fig. 3g). This is consistent with InsP3 and InsP5 each lacking the axial phosphate that is important for binding of InsP6. Importantly, mutation of Uba6 residues comprising the InsP6-binding pocket of Uba6 to the corresponding amino acids of Uba1 (K644E/S648L/L702W/K706H/K709T/Y710Q; hereafter referred to as Uba6_InsP6mut) completely abrogates the interaction between Uba6 and InsP6 as assessed using ITC (Fig. 3g and Supplementary Fig. 3c). Taken together, these results indicate that InsP6 binding is a feature specific to Uba6 and not Uba1 and that the basic binding pocket of Uba6 is also capable of binding to other signaling inositol phosphates such as InsP3 and InsP5. Lastly, the high affinity of Uba6 for InsP6 (~41 nM), coupled with intracellular concentrations of InsP6 in eukaryotic cells in the 10–100 µM range[43,44], suggests this binding has physiological significance. It also provides an explanation for the robust co-purification of InsP6 with Uba6 expressed in insect cells, even after several steps of purification.

## InsP6 modulates Uba6 activity and stability

Next, we examined whether InsP6 is capable of modulating the activity of Uba6 by determining pre-steady state kinetic parameters for Uba6-Ub and Uba6-FAT10 thioester bond formation in the presence and absence of 50 µM InsP6. This concentration was selected because it is the median cellular concentration reported in the literature. *E. coli*-derived Uba6 was used in these assays since as noted above, *E. coli* cannot synthesize InsP6. Interestingly, we found that while the Michaelis constant ($K_m$), is roughly the same for Ub and FAT10 in the

presence and absence of InsP6, the observed rate constant ($k_{obs}$) of Uba6-Ub and Uba6-FAT10 thioester intermediate formation is ~3-4 fold slower in the presence of InsP6 (Fig. 4a, b; Table 1; and Supplementary Figs. 4–6). Notably, the kinetic parameters for insect cell-derived Uba6 are similar to the *E. coli*-derived material in the presence of InsP6, indicating that the co-purified InsP6 in the insect cell-derived material similarly attenuates catalytic activity (Fig. 4a, b and Table 1).

To evaluate the effects of InsP6 on E1 activity further, dose-response experiments were conducted using E1-Ub and E1-FAT10 thioester formation assays using Uba6_WT, Uba6_InsP6mut, and Uba1_WT produced in *E. coli* with InsP6 concentrations ranging from 0 to 50 µM. The results show that InsP6 inhibits Uba6-Ub and Uba6-FAT10 with IC$_{50}$ values of 51 and 67 nM (Fig. 4c and Supplementary Fig. 7a), respectively, consistent with the $K_D$ of the Uba6/InsP6 interaction measured using ITC (41 nM). Importantly, InsP6 failed to inhibit Uba6_InsP6mut activation of both Ub and FAT10, and InsP6 failed to inhibit Uba1 activation of Ub (Fig. 4c). These results further demonstrate that the inhibitory effect of InsP6 is specific to Uba6 and is mediated through binding to the basic pocket on its SCCH domain. We also tested the ability of InsP3 and InsP5 to inhibit Uba6 activation of Ub and FAT10 using E1-Ub/FAT10 thioester formation assays (Fig. 4d and Supplementary Fig. 7b). These inositol phosphates exhibit significantly diminished inhibition of Uba6-Ub and Uba6-FAT10 thioester intermediate formation relative to InsP6 (Fig. 4d), which was surprising given that both InsP3 and InsP5 bind to Uba6 with high affinity ($K_D$ values of 248 and 73 nM, respectively; Fig. 3g). This result indicates an important role for the axial phosphate of InsP6 in modulating Uba6 activity.

Thermal shift assays were next conducted to determine whether inositol phosphate binding affects Uba6 stability using both full-length and standalone SCCH domain variants of Uba6 (Fig. 4e and Supplementary Fig. 8). The results indicate that InsP6 binding to the Uba6 SCCH domain derived from *E. coli* increases its melting temperature from ~43 to 59 °C, which is similar to the melting temperature of insect cell-derived protein (Fig. 4e). InsP3 and InsP5 increase the melting temperature of the Uba6 SCCH domain derived from *E. coli* from ~43 to 47 and 54 °C, respectively, whereas none of the inositol phosphates affect the stability of insect cell-derived protein (Fig. 4e). Lastly, none of the inositol phosphates affected the thermal stability of *E. coli*-derived Uba6_InsP6mut SCCH domain (Fig. 4e), which again demonstrates that the modulatory activity of inositol phosphates on Uba6 are mediated through binding to the basic pocket of the SCCH domain. Though the melting curves are more complicated due to the multidomain nature of Uba6, the trends described above for the Uba6 SCCH domain are similar in the context of full-length Uba6 (Supplementary Fig. 8).

## Mechanism of InsP6 modulation of Uba6 activity and stability

Analysis of our Uba6$^{OPEN}$/Ub-AMP and Uba6$^{CLOSED}$/Ub-AMP structures provides insights into the molecular mechanism by which InsP6 binding to Uba6 modulates its activity and stability. As noted above, many of the residues interacting with InsP6 in the Uba6$^{OPEN}$/Ub-AMP structure are located within regions of Uba6 that must become disordered (cys cap), adopt significantly different conformations (crossover loop and H18), or occupy significantly different locations due to SCCH domain alternation during the transition to the closed conformation for catalysis of thioester bond formation (Fig. 2a–c). This includes Lys628 from the crossover loop, Lys714 from the SCCH domain, and Lys800 and Lys810 from the cys cap, which together participates in an extensive network of hydrogen bonds and salt bridges with equatorial phosphates of InsP6 in the Uba6$^{OPEN}$/Ub-AMP structure (Fig. 5a, b). In the Uba6$^{CLOSED}$/Ub-AMP structure, all of these contacts to InsP6 are lost, and instead, Lys628 and Lys714 participate in salt bridges with Asp569 and Glu502 within the active site that contribute to the stabilization of the closed conformation (Fig. 5a, b).

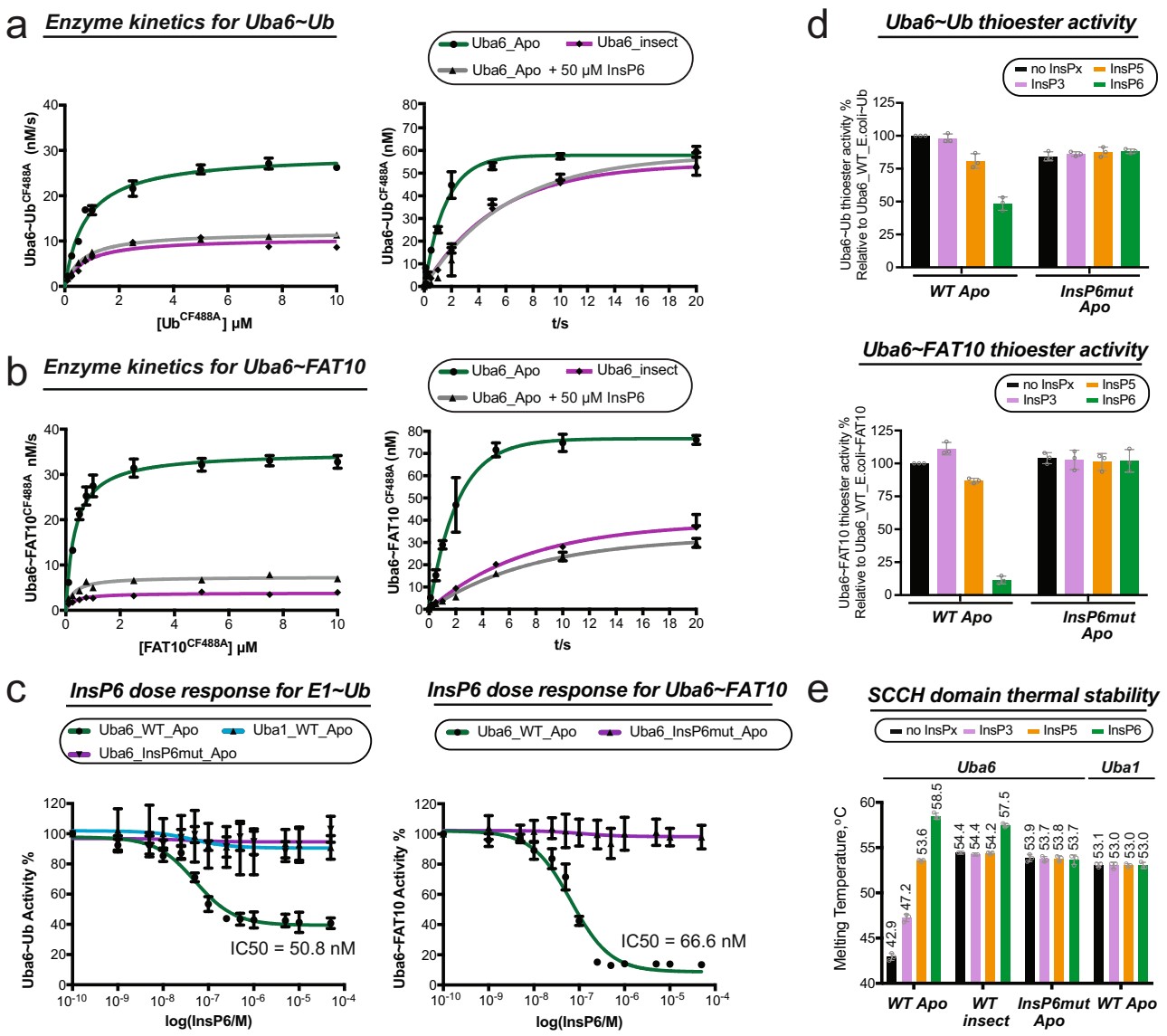

**Fig. 4 | InsP6 inhibits Uba6 E1-Ubl thioester formation activity and it enhances its stability. a, b** Kinetic curves of *E. coli*-derived Uba6 in the presence and absence of InsP6, and insect cell-derived Uba6, in Uba6-Ubl thioester activity assays. Throughout the figure, "Apo" labels refer to *E. coli*-derived material. Data are presented as mean values +/− SEM ($n = 3$ technical replicates). **c** IC50 curves for InsP6 inhibition of the Ub (*Left*) and FAT10 (*Right*) thioester formation activities of the indicated Uba6 and Uba1 variants. Data are presented as mean values +/− SEM ($n = 3$ technical replicates). **d** Effects of InsP6 and other inositol polyphosphates on the Uba6-Ub (Top) and Uba6-FAT10 (Bottom) thioester formation activities of the indicated Uba6 variants. Assays were conducted in the presence and absence of 100 μM InsPx(3, 5, or 6). Data are presented as mean values +/− SEM ($n = 3$ technical replicates) and displayed as percentages of the WT value with individual replicates shown as gray circles. **e** Melting temperatures of the indicated SCCH domain variants as determined by thermal shift assays in the presence and absence of 100 μM InsPx(3, 5, or 6). Data are presented as mean values +/− SEM ($n = 3$ technical replicates) with individual replicates shown as gray circles. Source data underlying Fig. 4a–e are provided as a Source data file.

This leads to the hypothesis that InsP6 binding to Uba6 inhibits its activity by stabilizing the open conformation of Uba6, thereby hindering the formation of the closed conformation necessary for thiolation. This occurs through InsP6-mediated stabilization of the cys cap that covers the catalytic cysteine and would clash with the adenylation domain if it remained ordered in the closed conformation (Fig. 5c), and stabilization of the crossover loop (including catalytic cysteine containing H18) in a conformation in which the catalytic cysteine is distant from the active site. Consistent with this hypothesis, we found that InsP6 modulatory activity is significantly diminished in K628A and K714D mutants of Uba6 in both Uba6-Ub thioester formation activity assays as well as in Uba6-Ub-AVSN crosslinking assays (Fig. 5d and Supplementary Fig. 9a, b). Ub-AVSN[45,46] harbors an electrophilic vinyl sulfonamide designed to bypass the adenylation half-reaction, thereby

specifically serving as a readout for the thiolation half reaction by covalently trapping the incoming E1 cysteine nucleophile[25,31]. Further, the $K_D$ of the K714D Uba6/InsP6 interaction increased ~30-fold (from 41 nM to 1.45 μM) (Supplementary Fig. 9c), in line with the loss of hydrogen bonds to the 6′ phosphate and long-range electrostatic interactions with the 5′ phosphate of InsP6. Also consistent with our hypothesis, both the E502A and D569A mutants exhibit severely diminished activity in both Uba6-Ub thioester formation and Uba6-Ub-AVSN crosslinking assays (Fig. 5d).

Altogether our data suggest that InsP6 mediates: (1) stabilization of flexible loops within the SCCH domain (cys cap, crossover loop), (2) stabilization of the SCCH domain in the open conformation, and by extension, rigidification of H1/H2, and g6 structural elements within the active site, and (3) stabilization of the globular SCCH domain itself

**Table 1 | Estimates of $K_m$ and $k_{obs}$ for Uba6~Ubl thioester formation assays**

| Uba6~Ub thioester reaction | $K_m$ (µM) | $V_{max}$ (nM s$^{-1}$) | $k_{obs}$ (s$^{-1}$) | $k_{obs}/K_m$ (µM$^{-1}$ s$^{-1}$) |
|---|---|---|---|---|
| Uba6_Apo | 0.77 ± 0.063 | 29.2 ± 0.652 | 0.63 ± 0.043 | 0.82 |
| Uba6_Apo + 50 µM InsP6 | 0.66 ± 0.030 | 12.0 ± 0.144 | 0.16 ± 0.021 | 0.24 |
| Uba6_WT | 0.71 ± 0.118 | 10.6 ± 0.471 | 0.19 ± 0.008 | 0.27 |
| **Uba6~FAT10 thioester reaction** | | | | |
| Uba6_Apo | 0.34 ± 0.028 | 35.0 ± 0.624 | 0.47 ± 0.036 | 1.4 |
| Uba6_Apo + 50 µM InsP6 | 0.27 ± 0.039 | 7.39 ± 0.217 | 0.13 ± 0.010 | 0.48 |
| Uba6_WT | 0.28 ± 0.038 | 3.83 ± 0.106 | 0.14 ± 0.009 | 0.5 |

The values for Uba6~Ubl formation were determined by plotting quantified signal versus time and fitting to the single exponential rise equation, $Y = A_0 \times (1 - \exp(-k_{obs} \times t))$.

by bridging the two lobes of the domain together (Fig. 3a). This leads us to hypothesize that these features collectively contribute to the significant increase in thermal stability of the Uba6/InsP6 complex compared to Uba6 alone (Fig. 4e and Supplementary Fig. 8). Indeed, this may explain the significantly higher yield of recombinant Uba6 when expressed in insect cells compared to expression in *E. coli*.

## Discussion

In this manuscript, we present structural snapshots of Uba6, which together reveal the molecular mechanisms by which adenylation and thiolation are catalyzed by this enzyme. The Uba6$^{OPEN}$/Ub-AMP structure represents the product complex of adenylation after the release of PPi/Mg$^{2+}$ and the Uba6$^{CLOSED}$/Ub-AMP structure represents the complex poised for catalysis of thiolation. A comparison of the open and closed structures reveals that SCCH domain alternation and active site remodeling disassemble the adenylation active site and reconfigure it for the thiolation reaction by swapping residues important for catalysis into the active site. Helix H14 of Uba6 is positioned to constitute the oxyanion hole of the active site by providing complementary positive electrostatic potential for stabilization of the transition state and tetrahedral intermediate formed during both adenylation and thioester bond formation.

While many of the salient mechanistic features of catalysis of adenylation and thiolation are conserved with other E1s such as Uba1[25] and the E1 for SUMO[31], our structures of Uba6 reveal an unexpected and unique feature of Uba6; namely the highly basic pocket on the SCCH domain that binds InsP6 which and serves to modulate both activity and stability of the enzyme. The proximity of the InsP6 binding site to structural elements that play a key role in SCCH domain alternation and active site remodeling provides an explanation for how InsP6 modulates Uba6 activity. Our data suggest that the InsP6-mediated reduction in Uba6 activity is due to the stabilization of structural elements that undergo conformational changes during the transition from the adenylation-competent to thiolation-competent state. Interestingly, our data suggest that these interactions contribute to the thermal stability of Uba6 through the same mechanisms. Our findings are similar to those for an allosteric SUMO E1 inhibitor called COH000[32]. Although its binding site is totally different compared to InsP6, COH000 also binds a pocket that is in proximity to structural elements key for SCCH domain alternation and active site remodeling and exploits these mechanistic features as part of its inhibitory mechanism[32]. The discovery of COH000 led to speculation that E1 enzymes might be allosterically regulated by naturally occurring cellular metabolites, and indeed, InsP6 is the first such example.

Notably, our data demonstrate that Uba6 can also bind other inositol phosphates. These include the second messengers InsP3 and InsP5, which bind with nanomolar affinity, although they do not inhibit the activity of Uba6 to the extent as InsP6 (Fig. 3g). By contrast, myo-inositol (which lacks phosphates) fails to bind with Uba6 with detectable affinity. In principle, more than 63 phosphorylated InsPs, can be generated by sequential phosphorylation of myo-inositol, and it remains to be seen how the incredible complexity in the diversity of inositol phosphates might interface with Uba6 function[47,48]. Compared to Uba6, Uba1 has a totally different structure and surface charge distribution at the region corresponding to the inositol polyphosphate binding site of Uba6 and fails to detectably interact with inositol polyphosphates (Fig. 3g). It is intriguing to speculate that the unique ability of Uba6 to interact with inositol polyphosphates may underlie the observation that only half of Uba6 molecules present at any time in the cell are activated with Ub/FAT10, compared to nearly all Uba1 molecules existing in the activated state[6,33]. The number of enzymes involved in Ub/FAT10 signaling demonstrated to interact with, and/or be regulated by inositol phosphates is steadily growing, with the first demonstrated examples being InsP5 and InsP6 regulation of cullin-RING E3 Ub ligase function in plants and humans[36–39]. This suggests there may be a subset of Ub/FAT10 signaling axes that are regulated by inositol phosphates governing distinct cellular processes. From a broader perspective, the Ub/FAT10 pathway is one of several signaling pathways that are now known to be regulated by high-affinity interactions with inositol phosphate second messengers and suggests that we are only beginning to understand the role these molecules play in cell biology[49,50].

Lastly, it is intriguing to speculate that other negatively charged cellular metabolites or macromolecules may engage the basic binding pocket of Uba6 under certain cellular contexts.

For instance, phosphoinositides, which harbor a variety of phosphorylated inositol headgroups, interact with a variety of cellular proteins in a regulated manner. These phosphorylated inositol headgroups serve as precursors for soluble inositol phosphates such as InsP3, InsP4, InsP5, InsP6 which we have demonstrated bind Uba6 with high affinity. This suggests that Uba6 may be capable of interacting with certain phosphoinositides, which, in addition to regulating its activity and stability, could serve as a mechanism to alter its subcellular localization. In this regard, it is noteworthy that roles for both Uba6 and phosphoinositides in autophagy have been reported[51,52]. Likewise, this study reinforces the possibility that other E1s can bind other naturally occurring cellular metabolites as a mechanism of regulation. Commensurate with this, we have previously identified a small pocket on the surface of the SCCH domain of Uba1 that is predicted to be a ligand-binding hotspot (Hotspot 3)[23]. The structure and electrostatics of Hotspot 3 in Uba1 differ from the InsP6-binding site of Uba6 (Fig. 3a–c), and therefore, any metabolite or macromolecule binding at this location would possess different structural and physicochemical properties compared to inositol phosphates.

While our work has illuminated the molecular mechanisms by which Uba6 catalyzes adenylation and thiolation, and also unexpectedly revealed a regulatory mechanism mediated by a cellular metabolite, much remains to be learned about this enzyme. This includes elucidating what role the binding of inositol phosphate plays in regulating the function of Uba6 in cells, and whether other cellular factors also bind the basic pocket on the SCCH domain to modulate its activity. Additional questions include defining the molecular rules governing the dual specificity of Uba6 for both Ub and FAT10, and elucidating the mechanism for the transfer of these Ub/Ubls to the Uba6 cognate E2 conjugating enzyme Ube2Z. In this regard, the InsP6 binding site of Uba6 overlaps with the predicted E2 binding surface of its SCCH domain and it will be exciting to determine whether InsP6 plays some role in E1-E2-Ub transthioesterification. Lastly, our Uba6/Ub-AMP structures, and in particular, differences with the human Uba1 active site, will provide a framework for drug discovery efforts targeting specifically this enzyme for therapeutic intervention in cancer and other diseases.

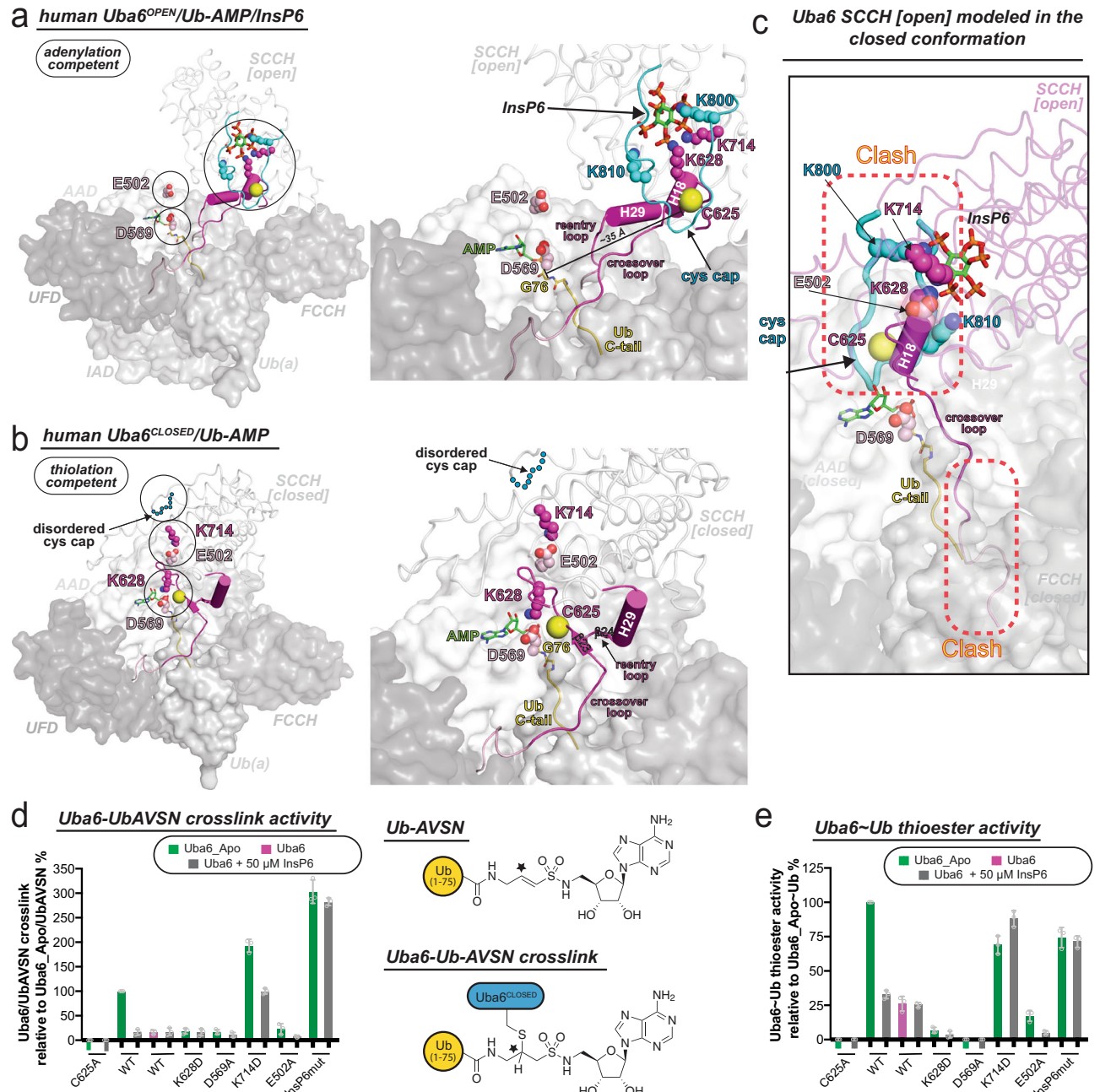

**Fig. 5 | Mechanism of InsP6 modulation of Uba6 activity and stability. a**, **b** The Uba6 open (**a**) and Uba6 closed (**b**) structures are shown as surface representations except for the SCCH domains, which are shown as loops. Elements that undergo conformational changes in the SCCH domain during transition from the adenylation to thiolation-competent state are shown as cartoons. Selected amino acids are shown as spheres. InsP6 and AMP are shown as sticks. Disordered cys cap is shown as cyan circles. **c** SCCH domain from the Uba6 open structure modeled in the closed conformation. SCCH open (shown as loops), was superimposed onto the SCCH domain of the Uba6 closed structure with Uba6 shown as surface representation. The steric clash between elements from SCCH [open] including cys cap, Helix 18, crossover loop, and reentry loop with Uba6 are highlighted by dashed red rectangles. **d** Structure–function analysis of Uba6 side chains depicted in **a** and **b** in

Ub-AVSN crosslinking assays that uncouple adenylation and thiolation and specifically assess thiolation activity (Left). Throughout the figure, "Apo" refers to *E. coli*-derived material. Schematic of Ub-AVSN and the Uba6-Ub-AVSN crosslink (Right). The electrophilic center attacked during reaction with Uba6 is indicated by a black star. Data are presented as mean values +/− SEM (*n* = 3 technical replicates) and displayed as percentages of the WT value with individual replicates shown as gray circles. **e** Structure–function analysis of Uba6 side chains depicted in **a** and **b** in Uba6-Ub thiolation assays that require both adenylation and thiolation. Data are presented as mean values +/− SEM (*n* = 3 technical replicates) and displayed as percentages of the WT value with individual replicates shown as gray circles. Source data underlying Fig. 5d, e are provided as a Source data file.

## Methods
### Cloning
For *E. coli* expression, the DNA fragment encoding human Uba6 residues 37–1052 was cloned into NdeI/NotI sites of vector pSMT3.2 with an N-terminal ULP1-cleavable SMT3 tag[53]. The DNA fragment

encoding human Uba6 SCCH domain (residues 615–892) or human Uba1 SCCH domain (residues 625–892) was synthesized (Gene Universal) and inserted into NdeI/NotI sites of vector pSMT3.4 with an N-terminal ULP1-cleavable SMT3 tag. The DNA fragment encoding human ubiquitin was inserted into NcoI/XhoI sites of vector

pET29NTEV with an N-terminal TEV-cleavable 6 × His tag. The DNA fragment encoding human FAT10 (C7T/C9T/C134L/C160S/C162S)[40] was synthesized and inserted into the BamHI/NotI sites of vector pGEX-6P1. Wheat ubiquitin with its seven lysines mutated to arginine (Ub$^{K7R}$) was prepared as previously described[54]. For insect cell expression, the DNA fragment encoding human Uba6 residues 37–1052 was cloned into NcoI/NotI sites of pFastBac HTB with an N-terminal TEV cleavable 6× His tag. The catalytic cysteine, Cys625, was mutated to alanine for crystallization. The DNA fragment encoding human Uba6 SCCH domain (residues 615–892) was cloned into BamH/NotI sites of pFastBac HTB with an N-terminal TEV cleavable 6× His tag. All point mutations were introduced using PCR-based site-directed mutagenesis. All constructs and point mutations were generated using the primer pairs described in Supplementary Table 2.

## Protein expression and purification
All proteins expressed in *E. coli* BL21 (DE3) Codon Plus cells (Agilent; Cat. No. 230280) were produced as described previously[23]. Briefly, large-scale cultures were grown at 37 °C in Luria Broth medium to A600 OD 2.0, and then placed in an ice bath for cold shock with the addition of 1.5% ethanol (v/v). After 30 min, the protein was induced by the addition of isopropyl-β-D-1-thioglactioside (IPTG) to a final concentration of 0.1 mM followed by shaking at 18 °C overnight. Bac-to-Bac Baculovirus Expression System was used to express human Uba6 in insect cells. High titer recombinant baculoviruses were used to infect *BTI-Tn-5B1–4* (Hi5) cells (ThermoFisher Scientific; Cat. No. B85502) at a cell density of $2 \times 10^6$ cells/ml cultured in Sf-900 II SFM medium (ThermoFisher Scientific). Cells were harvested after 48 h infection and stored at −80 °C before further use.

Bacterial- or insect cell-expressed Uba6 cells were harvested by centrifugation and lysed by sonication in lysis buffer (20 mM Tris HCl pH 8.0, 350 mM NaCl, 20 mM Imidazole, 0.5 mM TCEP), in the presence of DNase and Lysozyme. Cell lysate was centrifuged at $39,191 \times g$ for 30 min and the supernatant was then applied to Ni-NTA resin (QIAGEN), and the protein was eluted in buffer 20 mM Tris HCl pH 8.0, 350 mM NaCl, 250 mM Imidazole, 0.5 mM TCEP. The SMT3 tag was cleaved by adding ULP1 protease at a ratio of 1:2000 (w/w) and incubating overnight at 4 °C. The 6× His tag was cleaved by adding TEV protease at a ratio of 1:100 (w/w) and incubating overnight at 4 °C. After cleavage, the protein was subjected to Superdex 200 gel filtration (GE Healthcare) with buffer 20 mM Tris HCl pH 8.0, 350 mM NaCl, 0.5 mM TCEP, and later the target protein was pooled and subjected to MonoQ anion exchange column (GE Healthcare) with buffer (buffer A: 20 mM Tris HCl pH 8.0, 50 mM NaCl, 0.1 mM TCEP; buffer B: 20 mM Tris HCl pH 8.0, 1000 mM NaCl, 0.1 mM TCEP) for further purification. Uba6 SCCH domain and Uba1 SCCH domain were purified as described for Uba6, except for using a Superdex 75 column (GE Healthcare) instead of Superdex 200. FAT10 was purified by GST-affinity chromatography and Superdex 75 gel filtration (GE Healthcare). Human ubiquitin and wheat Ub$^{K7R}$ were purified by Ni-NTA affinity and Superdex 75 gel filtration (GE Healthcare). After purification, proteins were concentrated to 5–10 mg/ml, aliquoted, and snap frozen in liquid nitrogen.

## Crystallization and data collection
Uba6 C625A was purified as described above at a final concentration of 115 μM in 20 mM Tris HCl, pH 8.0, 150 mM NaCl, 0.5 mM TCEP. Wheat Ub$^{K7R}$ (230 μM), MgCl$_2$ (5 mM), and ATP (1 mM) were added prior to sparse-matrix screening in Intelli-Plate (Art Robbins Instruments) at 18 °C. Diffraction quality crystals of the Uba6/Ub complex were grown by mixing 1 μl of protein sample with 1 μl of crystallization buffer (100 mM MES pH 6.4, 140 mM NaF, 15% PEG3350). Crystals were flash-frozen in liquid nitrogen in a cryoprotectant composed of mother liquor supplemented with 30% PEG3350.

A complete X-ray diffraction data set from a single Uba6/Ub crystal was collected to a resolution of 2.25 Å at the Advanced Photon Source (Argonne, IL), NE-CAT beamline 22-ID-E. All data were indexed, integrated, and scaled using HKL2000[55]. The crystals belong to space group C2 with unit cell dimensions $a = 248.6$ Å, $b = 101.3$ Å, $c = 122.9$ Å, and $\alpha = 90°$, $\beta = 118°$, $\gamma = 90°$, with two hUba6/Ub complexes per asymmetric unit.

## Structure determination and refinement
The program Sculptor was used to generate a model of human Uba6 based on the coordinates of *H. sapiens* Uba1 (PDB: 6DC6)[23]. The program PHASER[56] was then used to find a molecular replacement solution using the coordinates for the individual domains (AAD/IAD, FCCH, SCCH, and UFD domains) of the Uba6 Sculptor model and wheat Ub (PDB: 4II2). The model was refined to R/R$_{free}$ values of 0.166/0.206 via iterative rounds of refinement and rebuilding using PHENIX[57] and COOT[58]. The initial round of refinement involved rigid body fitting of the AAD/IAD, FCCH, SCCH, and UFD of hUba6 and Ub. After the initial round of refinement, strong and continuous electron density at the C-termini of both molecules of Ub in the AU was consistent with the Ub-adenylate product, and AMP and the glycylphosphate linkage were subsequently built into the electron density after several additional rounds of model building and refinement. In the Uba6$^{OPEN}$/Ub-AMP structure, we also observed strong electron density within a pocket on the SCCH domain that we assigned as an InsP6 molecule based on the six-fold symmetry of the density and surrounding chemical environment. We manually placed an InsP6 into the density during the final rounds of refinement, which refined well. In the Uba6$^{CLOSED}$/Ub-AMP structure, with the exception of the position corresponding to the axial phosphate of InsP6, electron density for InsP6 is significantly weaker, and thus InsP6 was not modeled.

The final hUba6/Ub-AMP model contains amino acids 1–76 from both copies of Ub (chains B and D), amino acids 40–1050 from copy A of Uba6, and amino acids 59–798 and 819–1049 from copy C of Uba6. The model contains two AMP molecules from the chain B and D Ub-AMP intermediates, one InsP6 molecule associated with Uba6 chain A, and 846 water molecules. The final model has good geometry, with 97.1, 2.7, and 0.2% of residues in the favored, allowed, and disallowed regions of Ramachandran space, respectively. All molecular graphic representations of the structures were generated using PyMOL[59]. Structure alignments were performed using the program Superpose in CCP4 software suite[60].

## Isothermal titration calorimetry experiments (ITC)
ITC experiments were performed on an Affinity ITC (TA instruments) at 25 °C in buffer containing 20 mM HEPES pH 7.5, 150 mM NaCl, 0.5 mM TCEP. Aliquots (2.5 μl each) of 100 μM InsP3 (D-myo-Inositol 1,4,5-trisphosphate triammonium salt, Santa Cruz), InsP5 (D-myo-Inositol 1,3,4,5,6-pentakisphosphate pentapotassium salt, Enzo) or InsP6 (Phytic acid sodium salt hydrate, Sigma-Aldrich) were injected into a cell containing 10 μM full-length or SCCH domain of E1 proteins or Ubl. Twenty measurements were made and the data were analyzed using NanoAnalyze (TA instruments). Each experiment was conducted in triplicate.

## E1-Ubl activation assays
Gel-based E1 thioester formation assays were performed with 100 nM E1, 2.5 μM Ubl, 5 mM MgCl$_2$, 500 μM ATP, 137 mM NaCl, 2.7 mM KCl, 10 mM Na$_2$HPO$_4$, 1.8 mM KH$_2$PO$_4$ pH 7.4, 5% Glycerol and 0.1 mM TCEP at room temperature (RT). Reactions were initiated by adding ATP and were terminated by adding non-reducing urea SDS-PAGE buffer and subjected to 4–12% NuPAGE Bis-Tris gel (Life Technologies), 150 V constant for 60 min. The gels were stained with Sypro Ruby (BioRad) and visualized with a ChemiDoc MP (BioRad). Data quantification was conducted using densitometry in ImageJ 1.53 software and analyzed

using Prism 7.0a (GraphPad). Densitometry measurements were normalized as a percentage of the control WT assay on the same gel. Data are represented as an average of three technical replicates with ±standard deviation error bars. Unprocessed images of representative gels for all biochemical assays are provided in the Source data file.

### Inositol phosphate-mediated E1-Ubl inhibition assays

All E1-Ubl inhibition assays were performed by incubating 100 nM of the indicated E1 with 100 μM InsPx (IP3, IP5, or IP6) in a buffer containing 137 mM NaCl, 2.7 mM KCl, 10 mM $Na_2HPO_4$, 1.8 mM $KH_2PO_4$ pH 7.4, and 5 mM $MgCl_2$ for 5 min at RT. ATP and the corresponding Ubl were subsequently added to a final concentration of 500 and 2.5 μM, respectively. Reactions were incubated at RT for 2 s and terminated using non-reducing urea SDS-PAGE buffer. All samples were subjected to SDS-PAGE, stained, visualized, and quantified as described above for *E1-Ubl activation assays* except that densitometry measurements were normalized as a percentage of the control WT without InsPx assay on the same gel.

### E1-Ub-AVSN cross-linking assay

Crosslinking of wild-type and mutant constructs of E1 to Ub-AVSN was performed in a reaction mixture containing 100 nM E1, 500 nM Ub-AVSN, 137 mM NaCl, 2.7 mM KCl, 10 mM $Na_2HPO_4$, 1.8 mM $KH_2PO_4$ pH 7.4, 5% Glycerol and 0.5 mM TCEP at RT for 1 min and denatured in reducing SDS-PAGE buffer. Samples were subjected to SDS-PAGE, stained, visualized, and quantified as described above for E1-Ubl activation assays.

### Fluorescent labeling of Ubl

Fluorescent labeling of $Ubl^{Cys0}$ with CF488A maleimide (Biotium) was performed as recommended by the manufacturer. In Brief, 50 μM $Ubl^{Cys0}$ was incubated with 150 μM CF488A dye in conjugation buffer (20 mM HEPES pH 7.5, 150 mM NaCl, 0.5 mM TCEP) at RT overnight. Reactions were then quenched with 5 mM DTT and diluted to lower the concentration of NaCl to 50 mM. The resultant mixture was subjected to EnrichS (BioRad) column, and eluted with a linear gradient (for $Ub^{CF488A}$, 50 mM $NH_4Ac$, pH 4.51, 50–1000 mM NaCl; for $FAT10^{CF488A}$, 20 mM Bis-Tris, pH 6.5, 50–1000 mM NaCl). Purified $Ubl^{CF488A}$ was pooled and concentrated.

### Estimating the $K_m$ of Uba6~$Ubl^{CF488A}$ thioester formation

Reactions were carried out similar to the E1-Ubl thioester formation described above, with a few modifications. Reactions contained 100 nM E1, 0.1–10 μM $Ubl^{CF488A}$, 5 mM $MgCl_2$, 500 μM ATP, 137 mM NaCl, 2.7 mM KCl, 10 mM $Na_2HPO_4$, 1.8 mM $KH_2PO_4$ pH 7.4, 5% Glycerol and 0.1 mM TCEP for 2 s at room temperature (RT). Reactions were initiated by adding ATP and were terminated by adding non-reducing urea SDS-PAGE buffer and subjected to 4–12% NuPAGE Bis-Tris gel (Life Technologies), 150 V constant for 60 min. The gels were imaged on ChemiDoc MP (BioRad) with the Alex488 channel. All gels were imaged with a serial dilution of a known quantity of $FAT10^{CF488A}$ so that each gel contained a standard curve to convert band intensity to nanomoles of $FAT10^{CF488A}$ with ImageJ 1.53 software and analyzed using Prism 7.0a (GraphPad) by fitting data points into Michaelis–Menten model. Samples from each substrate concentration were performed in triplicate.

### Estimating the rate of Uba6~$Ubl^{CF488A}$ thioester formation ($K_{obs}$)

Reactions contained 100 nM E1, 2.5 μM $Ubl^{CF488A}$, 5 mM $MgCl_2$, 500 μM ATP, 137 mM NaCl, 2.7 mM KCl, 10 mM $Na_2HPO_4$, 1.8 mM $KH_2PO_4$ pH 7.4, 5% Glycerol and 0.1 mM TCEP for different time points (0–20 s) at room temperature (RT). For short time points (25 ms, 100 ms, 500 ms, 1 s), reactions were performed on QFM-4000 rapid-quench flow instrument. To initiate the reaction, 20 μl of a solution containing E1 and ATP in Syringe 1 was rapidly mixed with 20 μl of a solution

containing $Ubl^{CF488A}$ in Syringe 2 at RT, the reaction was then quenched with 20 μl 1 N HCl/3 × non-reducing Urea SDS-PAGE loading buffer. For long time points (2 s, 5 s, 10 s, 20 s), reactions were performed manually. Reactions were initiated by adding ATP and were terminated by adding non-reducing urea SDS-PAGE buffer. The samples were subjected to SDS-PAGE, stained, visualized, and quantified as described above for estimating the $K_m$ of Uba6-$Ubl^{CF488A}$ thioester formation.

### Dose response of InsP6 for E1-Ubl thioester formation

Gel-based E1-Ubl inhibition assays were performed by incubating 100 nM of the indicated E1 with 0–50 μM InsP6 in a buffer containing 137 mM NaCl, 2.7 mM KCl, 10 mM $Na_2HPO_4$, 1.8 mM $KH_2PO_4$ pH 7.4, and 5 mM $MgCl_2$ for 5 min at RT. ATP and the corresponding Ubl were subsequently added to a final concentration of 500 μM and 2.5 μM, respectively. Reactions were incubated at RT for 2 s and terminated using non-reducing urea SDS-PAGE buffer. Samples were subjected to SDS-PAGE, stained, and visualized as described above for *E1-Ubl activation assays*. Data quantification was conducted using densitometry in ImageJ 1.53 software and analyzed using Prism 7.0a (GraphPad) by fitting data points into 3 parameters log(inhibition)-response model. Data are represented as an average of three technical replicates with ±standard deviation error bars. Unprocessed images of representative gels for all biochemical assays are provided in the Source data file.

### Thermal shift assay

2 μg full-length or SCCH domain of E1 proteins was mixed with 5× SYPRO Orange dye (Thermo Fisher) in 20 mM HEPES pH 7.5, 150 mM NaCl, 0.25 mM TCEP to 20 μl in MicroAmp Fast optical 96-well reaction plate (Life Technologies). Each sample was prepared in triplicate. The 96-well was sealed with MicroAmp Optical Adhesive Film (Life Technologies) and then was placed into QuantStudio 3 qRT-RCR (Applied Biosystems). The melt curve method was run: continuous collection was selected; 25 °C 2 min, 1.6 °C/s; 0.05 °C/s ramp; 95 °C 2 min. Data was saved for further constructing melt curves and determining melting temperature by Prism 7.0a (GraphPad).

### Reporting summary

Further information on research design is available in the Nature Research Reporting Summary linked to this article.

## Data availability

Atomic coordinates and structure factors are deposited in the Protein Data Bank (PDB) with accession code 7SOL. Previously published structural data used from the PDB are listed below: PDB: 6DC6, PDB: 4II2, PDB: 6GF2. Source data are provided with this paper.

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

## Acknowledgements
The authors thank Patrick Sung and Miklos Bekes for critically reading the manuscript. The X-ray diffraction data were collected on beamline NE-CAT 24-ID-E at the Advanced Photon Source, Argonne National Laboratory. This work is based upon research conducted at the Northeastern Collaborative Access Team beamlines, which are funded by the National Institute of General Medical Sciences from the National Institutes of Health (P30 GM124165). The Eiger 16M detector on 24-ID-E is funded by an NIH-ORIP HEI grant (S10OD021527). This research used resources of the Advanced Photon Source, a U.S. Department of Energy (DOE) Office of Science User Facility operated for the DOE Office of Science by Argonne National Laboratory under Contract No. DE-AC02-06CH11357. Research reported in this publication was supported by the NIH R01 GM115568, R01 GM128731, and CPRIT RR200030 (S.K.O.). This research utilized resources of the Structural Biology Core Facilities, part of the Institutional Research Cores at the University of Texas Health Science Center at San Antonio supported by the Office of the Vice President for Research and the Mays Cancer Center Drug Discovery and Structural Biology Shared Resource (NIH P30 CA054174). The Rigaku HyPix-6000HE Detector, Universal Goniometer, and VariMax-VHF Optic instrumentation in the Structural Biology Core Facilities are funded by NIH-ORIP SIG Grant S10OD030374. The content of this study is solely the responsibility of the authors and does not necessarily represent the official views of the NIH.

## Author contributions
Protein purification was conducted by F.G., L.Y., D.N., A.N., and P.S.B. F.E.O synthesized, purified, and validated activity-based probes. Structural experiments and analysis were conducted by F.G., L.Y., Z.L., and S.K.O. L.Y. and F.G. conducted biochemical and biophysical assays. E.V.W., K.E.C., L.J., F.C.A., B.O., K.M.W., and C.D. assisted with experimental design and data interpretation. The figures and manuscript were prepared by L.Y., F.G., and S.K.O. with input from all authors.

## Competing interests
F.E.O. declares competing financial interests as co-founder and shareholder of UbiQ Bio BV. All other authors declare no competing interests.
