## [Peer Review File · Nature Communications]

REVIEWER COMMENTS

Reviewer #1 (Remarks to the Author):

The manuscript by Yuan and Gao et al focuses on deciphering how Uba6 activates ubiquitin (and other Ubls). They present nice structural work based on their ability to obtain two different conformations of the complex in the crystals. In essence the structural data is consistent with observations that have been made for the related protein Uba1, and is in some ways confirmatory. However, in addition to trapping two states they observe the presence of the cofactor Inositide6P in the Open adenylation competent conformation. This is an interesting observation that they go on to investigate further, showing that binding of Ins6p appears to inhibit activity of the enzyme.

The manuscript contains considerable data and will be of interest to those in the field.

Major points:

- 1) Analysis of the samples by mass spectrometry might have provided insight into the ratio of InsP molecules bound?
- 2) It would be interesting to know if disrupting the Ins6P binding site altered the function of Uba6 in cells.
- 3) Page 4, line 86 – the authors should consider whether including the TRAF2 example is helpful as there is considerable doubt as to whether TRAF2 is an active E3 and the role of sphingosine has been disputed (e.g. Etemadi et al, 2015, eLife)
- 4) Line 172 – ‘bending’ of residues maybe better rephrased as a residue does not generally bend
- 5) There are many very long sentences e.g. more than 5 lines (examples are line 271-277; line 306-311; line 393-397). In general these need to be reorganised or split into two sentences as it is difficult to connect all the points. Second sentence in the Discussion is also a pretty tough read.
- 6) In general the manuscript needs a good edit and many sentences might be helped by the addition of commas.
- 7) Figure 1a – adenylation not adeylation
- 8) Figure 2a – overlay at right needs the colours stated. In overlays, it may be better to consistently use grey for one state.
- 9) Some small pink labels i.e. in Figure 2b are hard to read
- 10) Fig S2b – consistently refer to the closed state when referring to ‘thioester competent state’

11) Figure 3f – highlighting the Uba1 sequences with a pale grey background may emphasise the lack of conservation.

12) Their needs to be some comment on the reliability of the ITC data, i.e. was it repeated?

Reviewer #2 (Remarks to the Author):

Manuscript review for Nature communications (APRIL 6-2022)

Crystal Structures reveal catalytic and regulatory mechanisms of the dual-specific Ubiquitin/FAT10 enzyme Uba6

SK Olsen lab.

This manuscript reports the first structure of Uba6, an E1 enzyme with dual specificity for Ubiquitin and FAT10. Two distinct arrangements of a binary Uba6/Ub complex were observed in a single protein crystal. One protein complex has Ub-AMP positioned in the Uba6 adenylation site with the second catalytic cysteine half (SCCH) domain in the “open” conformation. Strikingly, the other protein complex represents the “closed” conformation and differs via a series of Uba6 structural rearrangements where the Ub-AMP is poised for transfer to the juxtapositioned SCCH and catalytic cysteine site. As such, a single protein crystal contains two snapshots of the Uba6/Ub-AMP complex representing distinct states in the E1 catalytic cycle.

While the overall E1 domain organization is highly related to Ubiquitin-specific Uba1, this dual-specific Uba6 contains a unique basic pocket on the SCCH domain that surprisingly binds inositol hexakisphosphate (IP6) in the open state but not (convincingly) in the closed state.

The authors claim that allosteric regulation by IP6 modulates conversion of Uba6 to the closed state and thus limits thioester formation with Ub/FAT10 and subsequent transfer to an E2 enzyme.

The allosteric mechanism was probed by mutagenesis of the IP6 pocket and by functional assays on Uba6 purified from bacteria in the absence of IP6. Interaction studies revealed IP6 bound Uba6 with ~40 nM affinity but was not detectable for the SCCH pocket mutant or Uba1 in comparison.

One caveat of these findings is that all reported ITC binding studies were conducted in the absence of Ubiquitin and ATP, thus were probing the affinity of IP6 for the Uba6-apo state and thus not the Ub-AMP bound state. However, this result, and the additional thermal melt assays, support the claim that IP6 was indeed the ligand bound in their crystal structure.

Both Ub- and FAT10-thioester formation with Uba6 was 3-4 fold slower in the presence of IP6 while the SCCH pocket mutant was not impacted under the same conditions. This effect was more pronounced than with related metabolites of IP3 and IP5 supporting the importance of the 6' phosphate in IP6 for the observed effect. A point mutation of K714D restores Ub thioester formation and Cys625 reactivity to a Ub-covalent analogue.

While the structure panels in Figure 5 do not make the role of K714 in IP6 interaction easy to visualize, the conclusions drawn suggest that disrupting the IP6 interaction at the 6' phosphate position is coupled to releasing/disordering the cysteine cap for thioester formation. Given the central importance of K714 to the proposed IP6 allosteric mechanism, the authors should include the results of binding data with this single point mutation.

Overall, this is a compelling structural and functional investigation of Uba6. The description of an allosteric role for IP6 in Uba6 function will be of interest to the broad readership of Nature Communications.

Points the authors may wish to consider:

- 1) Given the dual Ub- and FAT10-specificity of Uba6, the authors do not provide any analysis or comparison of the Uba6-Ub interface with a modeled FAT10 interaction. I think that this would be a fun topic to elaborate on.

- 2) Descriptions of differences between open and closed structural states were somewhat repetitive. Despite this comment, what aspects of the Ub-AMP interface with Uba6 remained consistent, i.e. is the Ub-AMP repositioned between distinct sites on the IAD-AAD to facilitate thioester formation or is a stable Uba6-Ub platform retained for the catalytic site transition? [This is an example where text and figures are less ideal than being able to view the protein complex using graphics software]

3) For clarification, is dissociation of pyrophosphate from the Ub-adenylation site a requirement for SCCH transition to the closed thioester-competent state of Uba6? Is this the point of Supplementary Figure 2c?

4) The authors propose that IP6 interaction in the SCCH pocket is inconsistent with the closed state due to diminished electron density and structural changes of the Cys cap and other local features that presumably diminish the binding affinity. Is dissociation of IP6 absolutely required due to any presumable steric clashes within the closed state? In other words, does Uba6 need to transition to a IP6 bound vs IP6-free state for the catalytic cycle to progress?

5) How does the UFD conformation compare with the previously described Uba1-E1/E2/Ub complex that represents the E2 trans-thiolation step. In addition to preventing a closed conformation for Ub-thioester formation, is the IP6-engaged open conformation also incompatible with engagement of an E2 by the UFD domain of Uba6? How is UFD conformational change linked to allosteric engagement by a bound IP6 molecule?

Minor points:

1) Typo in figure 1a label for adenylation. Uba6OPEN text difficult to read on blue domain cartoon.

Overall an excellent manuscript.

Reviewers' comments:

Reviewer #1 (Remarks to the Author):

The manuscript by Yuan and Gao et al focuses on deciphering how Uba6 activates ubiquitin (and other UbIs). They present nice structural work based on their ability to obtain two different conformations of the complex in the crystals. In essence the structural data is consistent with observations that have been made for the related protein Uba1, and is in some ways confirmatory. However, in addition to trapping two states they observe the presence of the cofactor Inositide6P in the Open adenylation competent conformation. This is an interesting observation that they go on to investigate further, showing that binding of Ins6p appears to inhibit activity of the enzyme.

The manuscript contains considerable data and will be of interest to those in the field.

We are grateful to the reviewer for their positive comments and thoughtful analysis of our study.

Major points:

1) *Analysis of the samples by mass spectrometry might have provided insight into the ratio of InsP6 molecules bound?*

We agree, and early in our studies we attempted to use mass spectrometry to validate the presence of InsP6 in our sample, as well as to address the ratio of InsP6 molecules bound. We learned that polyanionic molecules, such as InsP6, are notoriously difficult to analyze by mass spectrometry due to low retention in reverse-phase liquid chromatography and the presence of salts in the buffer which form adducts. Formation of these adducts result in multiple analyte-adduct ions, complicating the recorded spectra and reducing ion yields. We nevertheless produced the very large amount of protein necessary for initial studies but did not make good progress. In the meantime, we obtained data which was consistent with the presence of InsP6 using structure-function studies (Figs 3g, 4a-e, 5d-e). This included ITC experiments which yielded an N-value of nearly 1 for the WT Uba6/InsP6 interaction, suggesting a nearly 1:1 molar ratio of Uba6:InsP6. We thus paused our mass spectrometry efforts at this stage. With respect to the structure, although we did not build an InsP6 molecule into the 'closed' structure, we did note that there is electron density corresponding to where the axial phosphate of InsP6 is expected to be based on the 'open' structure (Fig. 3e). The rest of this putative InsP6 molecule is poorly ordered. Although speculative, the data suggests that InsP6 is bound in in the 'closed' conformation, but that aside from the axial phosphate, is poorly ordered due to loss of contacts between Uba6 and InsP6 that accompany the conversion from open to closed state.

2) *It would be interesting to know if disrupting the Ins6P binding site altered the function of Uba6 in cells.*

We completely agree with this. During revision, we initiated collaborative studies to study the role of InsP6 in Uba6 function in cells, though it has become apparent that these studies will be much more involved than could be completed during the time frame of the revision. This is a very interesting and potentially exciting question and we plan to continue these studies for inclusion in a future manuscript.

3) *Page 4, line 86 – the authors should consider whether including the TRAF2 example is helpful as there is considerable doubt as to whether TRAF2 is an active E3 and the role of sphingosine has been disputed (e.g. Etemadi et al, 2015, eLife)*

We thank the reviewer for pointing this out and have removed this citation and the accompanying text from the revised manuscript.

4) *Line 172 – 'bending' of residues maybe better rephrased as a residue does not generally bend.*

Thank you for pointing this out. We have rephrased this sentence (lines 183-185): "SCCH domain alternation is accompanied by a bending of the crossover loop between residues Arg615 to Pro623 and an orthogonal bending of the reentry loop between residues Gly888 to Ala893".

5) *There are many very long sentences e.g. more than 5 lines (examples are line 271-277; line 306-311; line 393-397). In general these need to be reorganised or split into two sentences as it is difficult to connect all the points. Second sentence in the Discussion is also a pretty tough read.*

We thank the reviewer for their constructive criticism and have thoroughly edited the entire manuscript with a particular emphasis on the points raised here.

6) *In general the manuscript needs a good edit and many sentences might be helped by the addition of commas.*

We appreciate the suggestions and have thoroughly edited the entire manuscript as noted above.

7) *Figure 1a – adenylation not adeylation*

Corrected, thank you.

8) *Figure 2a – overlay at right needs the colours stated. In overlays, it may be better to consistently use grey for one state.*

We have added a legend and colored the closed conformer of Uba6 in gray in the overlay, as well as the standalone panel.

9) *Some small pink labels i.e. in Figure 2b are hard to read*

We have increased the size of the labels, made them darker, and added a black outline.

10) *Fig S2b – consistently refer to the closed state when referring to ‘thioester competent state’*

We have relabeled this figure accordingly.

11) *Figure 3f – highlighting the Uba1 sequences with a pale grey background may emphasise the lack of conservation.*

We appreciate the suggestion and have highlighted Uba1 sequences with a gray background.

12) *Their needs to be some comment on the reliability of the ITC data, i.e. was it repeated?*

Apologies for this confusion. We conducted the ITC experiments in triplicate. We have further emphasized this in the Methods section and in the legends for Figure 3f and Supplementary Figure 5c.

Reviewer #2 (Remarks to the Author):

Manuscript review for Nature communications (APRIL 6-2022)

Crystal Structures reveal catalytic and regulatory mechanisms of the dual-specific Ubiquitin/FAT10 enzyme Uba6
SK Olsen lab.

This manuscript reports the first structure of Uba6, an E1 enzyme with dual specificity for Ubiquitin and FAT10. Two distinct arrangements of a binary Uba6/Ub complex were observed in a single protein crystal. One protein complex has Ub-AMP positioned in the Uba6 adenylation site with the second catalytic cysteine half (SCCH) domain in the “open” conformation. Strikingly, the other protein complex represents the “closed” conformation and differs via a series of Uba6 structural rearrangements where the Ub-AMP is poised for transfer to the juxtapositioned SCCH and catalytic cysteine site. As such, a single protein crystal contains two snapshots of the Uba6/Ub-AMP complex representing distinct states in the E1 catalytic cycle. While the overall E1 domain organization is highly related to Ubiquitin-specific Uba1, this dual-specific Uba6 contains a unique basic pocket on the SCCH domain that surprisingly binds inositol hexakisphosphate (IP6) in the open state but not (convincingly) in the closed state. The authors claim that allosteric regulation by IP6 modulates conversion of Uba6 to the closed state and thus limits thioester formation with Ub/FAT10 and subsequent transfer to an E2 enzyme.

The allosteric mechanism was probed by mutagenesis of the IP6 pocket and by functional assays on Uba6 purified from bacteria in the absence of IP6. Interaction studies revealed IP6 bound Uba6 with ~40 nM affinity but was not detectable for the SCCH pocket mutant or Uba1 in comparison. One caveat of these findings is that all reported ITC binding studies were conducted in the absence of Ubiquitin and ATP, thus were probing the affinity of IP6 for the Uba6-apo state and thus not the Ub-AMP bound state. However, this result, and the

additional thermal melt assays, support the claim that IP6 was indeed the ligand bound in their crystal structure. Both Ub- and FAT10-thioester formation with Uba6 was 3-4 fold slower in the presence of IP6 while the SCCH pocket mutant was not impacted under the same conditions. This effect was more pronounced than with related metabolites of IP3 and IP5 supporting the importance of the 6' phosphate in IP6 for the observed effect. A point mutation of K714D restores Ub thioester formation and Cys625 reactivity to a Ub-covalent analogue.

While the structure panels in Figure 5 do not make the role of K714 in IP6 interaction easy to visualize, the conclusions drawn suggest that disrupting the IP6 interaction at the 6' phosphate position is coupled to releasing/disordering the cysteine cap for thioester formation. Given the central importance of K714 to the proposed IP6 allosteric mechanism, the authors should include the results of binding data with this single point mutation.

Overall, this is a compelling structural and functional investigation of Uba6. The description of an allosteric role for IP6 in Uba6 function will be of interest to the broad readership of Nature Communications.

We appreciate the positive comments and insightful analysis of our study. As suggested, we have conducted ITC experiments to probe the interaction between InsP6 and the Uba6 K714D mutant, and now include this data in Supplementary Fig. 5c. The K_D of the interaction between the Uba6 K714D mutant and InsP6 increased by ~30 fold compared to Uba6 WT (from ~40 nM to 1.35 μ M). This is consistent with the loss of hydrogen bonds to the 6' phosphate and long-range electrostatic interactions with the 5' phosphate of InsP6 and the introduction of charge repulsion between the negatively charged K714D side chain and equatorial phosphates of InsP6.

Points the authors may wish to consider:

1) Given the dual Ub- and FAT10-specificity of Uba6, the authors do not provide any analysis or comparison of the Uba6-Ub interface with a modeled FAT10 interaction. I think that this would be a fun topic to elaborate on.

Originally, we did not wish to overspeculate on the matter of Uba6 promiscuity/FAT10 specificity as this remains an area of active investigation in the lab, but we agree with the reviewer that this is an important issue should be addressed at some level in this manuscript. Thus, we created a Uba6/FAT10 model and performed a comparative analysis with the Uba6/Ub structure. Please see added text (lines 154-175) and Supplementary Figs. 1d,e, and f for the results of this analysis. While speculative, the results of our analysis provides some interesting insights into promiscuity of Uba6 for Ub and FAT10, though a deeper understanding will require a *bona fide* structure of a Uba6 complex. Text added to the manuscript:

“Overall, the Uba6/Ub interaction is similar to that of Uba1/Ub and to gain insights into the dual specificity of Uba6 for Ub and FAT10, we created a Uba6/FAT10 model by docking FAT10 (PDB: 6GF2) onto Ub from the Uba6/Ub structure (Supplementary Fig. 1d,e). The Uba6/FAT10 model and sequence analysis reveal that FAT10 harbors only 31% identity at Ub positions that interact with Uba6 (Supplementary Fig. 1f). Among the many regions of divergence between Ub and FAT10, the Ile44 hydrophobic patch of Ub (Leu8/Ile44/Val70), which is known to be crucial for Uba1 activation of Ub and participates in a similar network of hydrophobic interactions in the Uba6/Ub structure, is not conserved in FAT10 (Supplementary Fig. 1d,e,f). In FAT10, the region corresponding to the Ub Ile44 patch has diverged to be much more polar and comprises Ser95, Thr132, and Ala159. Thus, the same network of functionally important hydrophobic contacts that Ub participates in with Uba1 and Uba6 is not possible for FAT10. In addition to differences at many other Ub residues that interact with Uba6 in the Uba6/Ub structure, FAT10 has a two-residue insertion in the β 1- β 2 loop which is in proximity to the AAD of Uba6 where it may participate in unique interactions (Supplementary Fig. 1d,e,f) and a novel ‘CYCI’ motif at the C-terminus that plays a role in specificity. Lastly, FAT10 differs from Ub in that it harbors two tandem Ub-like domains (NTD and CTD) connected by a linker region (Supplementary Fig. 1e) that previous studies have demonstrated is crucial for activation of FAT10 by Uba6 40 despite being situated quite far from the surface of the enzyme. . . Precisely how the aforementioned differences in the FAT10 CTD affect its mode of binding to Uba6, how the NTD-CTD linker and possibly the NTD itself might play a role in Uba6 binding, and whether these differences may compensate for the lack of the hydrophobic Ile44 patch in FAT10 will await determination of a Uba6/FAT10 structure.”

2) Descriptions of differences between open and closed structural states were somewhat repetitive. Despite this comment, what aspects of the Ub-AMP interface with Uba6 remained consistent, i.e. is the Ub-AMP repositioned between distinct sites on the IAD-AAD to facilitate thioester formation or is a stable Uba6-Ub platform retained for the catalytic site transition? [This is an example where text and figures are less ideal than being able to view the protein complex using graphics software]. 3) For clarification, is dissociation of

pyrophosphate from the Ub-adenylation site a requirement for SCCH transition to the closed thioester-competent state of Uba6? Is this the point of Supplementary Figure 2c?

We appreciate the comment, and as noted above, we substantially edited the manuscript to try to reduce repetition and improve overall clarity. We will combine our responses to points 2 and 3, as they are related. With respect to Ub-AMP, the Uba6^{OPEN}/Ub-AMP structure represents the product of the adenylation reaction, after pyrophosphate has been released from the active site. The point of Supplementary Figure 2c was to model a substrate complex (e.g. before adenylation and pyrophosphate release) by docking ATP/Mg²⁺ onto the Uba6^{OPEN}/Ub-AMP structure. This model demonstrates that conserved elements of the Uba6 active site important for catalysis of adenylation are configured for contacts to the β and γ - phosphates of ATP/Mg²⁺ (as well as the pyrophosphate leaving group), consistent with our naming of this conformer as 'adenylation-competent'. This includes residues Arg22, Arg508, and Gln509 from the H1/H2 and g6 regions of Uba6 which are displaced from the active site during transition from the open adenylation-competent conformation to the closed thioester-competent conformation. Dissociation of pyrophosphate from the active site is indeed a requirement for SCCH transition to the closed thioester-competent state, as the loss of contacts to H1/H2 and g6 residues facilitates their displacement from the active site which is necessary for SCCH domain closure. If H1/H2 and g6 elements were not displaced from the active site, they would engage in major steric clashes with the SCCH domain in the closed, thiolation-competent conformation.

With that said, the interface between Ub-AMP and Uba6 is retained in both our Uba6^{OPEN}/Ub-AMP and Uba6^{CLOSED}/Ub-AMP structures. Any differences that occur during the transition from adenylation to thiolation competent conformers of Uba6 are between the β and γ - phosphates of ATP/pyrophosphate and the H1/H2/g6 regions of Uba6.

We appreciate the reviewer bringing this potential point of confusion to light and we have extensively edited the labels and legend of Supplementary Fig. 2 and the main text to clarify these issues.

4) The authors propose that IP6 interaction in the SCCH pocket is inconsistent with the closed state due to diminished electron density and structural changes of the Cys cap and other local features that presumably diminish the binding affinity. Is dissociation of IP6 absolutely required due to any presumable steric clashes within the closed state? In other words, does Uba6 need to transition to a IP6 bound vs IP6-free state for the catalytic cycle to progress?

This is a great question that is related to Reviewer 1, point 1. To gain insights into the issue of dissociation of InsP6 in the closed state, we docked InsP6 onto the SCCH domain of the Uba6^{CLOSED}/Ub-AMP structures which we now include in new Supplementary Fig. 3b. While there are some steric clashes between the modeled InsP6 and Glu502, Lys503, Asn505 from the g6 region of Uba6, there are no major backbone clashes. As noted above, although we did not build an InsP6 molecule into the 'closed' structure, we did note that there is electron density corresponding to where the axial phosphate of InsP6 is expected to be based on the 'open' structure (Fig. 3e). The rest of this putative InsP6 molecule is poorly ordered. Although speculative, the data (along with the N-value of nearly 1 for the WT Uba6 interaction with InsP6) suggests that InsP6 is indeed bound to Uba6 in the 'closed' conformation. However, aside from the axial phosphate, this InsP6 molecule is poorly ordered due to loss of contacts between Uba6 and InsP6 that accompany the conversion from open to closed state. This suggests that cycling of bound/unbound InsP6 is not required for the catalytic cycle to progress but that local conformational changes are. Please see lines 305-310 of the revised manuscript for summarized description of this.

5) How does the UFD conformation compare with the previously described Uba1-E1/E2/Ub complex that represents the E2 trans-thiolation step. In addition to preventing a closed conformation for Ub-thioester formation, is the IP6-engaged open conformation also incompatible with engagement of an E2 by the UFD domain of Uba6? How is UFD conformational change linked to allosteric engagement by a bound IP6 molecule?

Thank you for this insightful question. We observe the UFD in a 'distal' conformation in the Uba6^{OPEN}/Ub-AMP structure and in the 'proximal' conformation in the Uba6^{CLOSED}/Ub-AMP structure. Uba1-E2/Ub transthioesterification requires transition of the UFD from a 'distal' to 'proximal' conformation that brings the E1 and E2 active sites in proximity to each other to facilitate catalysis. Contacts between the E2 and SCCH domain that promote catalysis occur during this process. Intriguingly, the InsP6 molecule bound to the SCCH

domain of Uba6 is in on the surface of the SCCH domain that is predicted to be in proximity to the E2 in a Uba6/E2/Ub complex. Interestingly, the InsP6 binding site on the SCCH domain of Uba6 overlaps with its predicted E2 bind site. An intriguing question is whether InsP6 may play some role in E1-E2 transtioesterification, and if so, what that role might be. We are actively conducting experiments to address these questions and have highlighted this interesting point in the Discussion of the revised manuscript (lines 499-501).

Minor points:

1) *Typo in figure 1a label for adenylation. Uba6OPEN text difficult to read on blue domain cartoon.*

Thank you, we have fixed these.

Overall an excellent manuscript.

Thank you.